# Alpine hillslope failure in the western US: Insights from the Chaos Canyon landslide, Rocky Mountain National Park USA

Matthew C. Morriss[1], Benjamin Lehmann[2,3], Benjamin Campforts[2,4], George Brencher[5],
Brianna Rick[6,7], Leif Anderson[8], Alexander L. Handwerger[9,10], Irina Overeem[2], and Jeffrey Moore[8]

[1]Earth Sciences Department, University of Oregon Eugene, OR, USA 97403
[2]Institute of Arctic and Alpine Research, Univesrity of Colorado, Boulder, CO, USA
[3]Univ. Grenoble Alpes, Univ. Savoie Mont Blanc, CNRS, IRD, Univ. Gustave Eiffel, ISTERRE 38000 Grenoble, France
[4]Department of Earth Sciences, VU University Amsterdam, Amsterdam, 1081HV, the Netherlands
[5]Civil and Environmental Engineering, University of Washington, Seattle, WA, USA
[6]Department of Geosciences, Colorado State University, Fort Collins, CO 80523, USA
[7]Alaska Climate Adaptation Science Center, Fairbanks, AK, 99775, USA
[8]Department of Geology and Geophysics, Salt Lake City, UT, USA
[9]Jet Propulsion Laboratory, California Institute of Technology, Pasadena, 91109, USA
[10]Joint Institute for Regional Earth System Science and Engineering, University of California, Los Angeles, Los Angeles, CA 90095, USA

**Correspondence:** Matthew Morriss, PhD (matthew.c.morriss@gmail.com)

**Abstract.** The Chaos Canyon landslide, which collapsed on the afternoon of June 28th, 2022 in Rocky Mountain National Park presents an opportunity to evaluate instabilities within alpine regions faced with a warming and dynamic climate. Video documentation of the landslide was captured by several eyewitnesses and motivated a rapid field campaign. Initial estimates put the failure area at 66,630 m², with an average elevation of 3,555 m above sea level. We undertook an investigation of previous movement of this landslide, measured the volume of material involved, evaluated the potential presence of interstitial ice/snow within the failed deposit, and examined potential climatological impacts on the collapse of the slope. Satellite radar and optical measurements were used to measure deformation of the landslide in the five years leading up to collapse. From 2017 to 2019, the landslide moved ∼5 m yr⁻¹, accelerating to 17 m yr⁻¹ in 2019. Movement took place through both internal deformation and basal sliding. Climate analysis reveals the collapse took place during peak snowmelt, and 2022 followed 10 years of higher than average positive degree day sums. We also made use of slope stability modeling to test what factors controlled the stability of the area. Models indicate even a small increase in the water table reduces the Factor of Safety to <1, leading to failure. We posit that a combination of permafrost thaw from increasing average temperatures, progressive weakening of the basal shear zone from several years of movement, and increase in pore-fluid pressure from snowmelt led to the June 28th collapse. Material volumes were estimated using Structure from Motion (SfM) models incorporating photographs from two field expeditions on July 8th, 2022 – 10 days after the slide. Detailed mapping and SfM models indicate ∼ 1,258,000 ± 150,000 m³ of material was deposited at the slide toe and ∼1,340,000 ± 133,000 m³ of material was evacuated from the source area. The Chaos Canyon landslide may be representative of future dynamic alpine topography, wherein slope failures becomes more common in a warming climate.

## 1 Introduction

The Chaos Canyon collapse took place during a sunny early-summer day at 3:31 PM local time on June 28th, 2022 in Rocky Mountain National Park (RMNP), Colorado, USA (Fig. 1). The event was first reported through social media from bystanders situated at Lake Haiyaha (Fig. 1) and directly beneath the slide. These social media posts resulted in a rapid response from Earth scientists who initially sought to investigate: 1) if the failure was spontaneous or part of a longer lived behavior; 2) the mechanisms contributing to this collapse - such as a landslide, the collapse of a rock glacier, and the the role of cryogenic processes in general - perhaps related to climate, and 3) the amount of material mobilized by the collapse. More broadly, this investigation developed through a desire to understand whether this failure is part of the global trend toward cryosphere instability and the degradation of permafrost conditions driven by climate change (Geertsema et al., 2022; Patton et al., 2019). Mass movements in alpine regions have been documented with increasing frequency in the European Alps, Canada, and Alaska (Dai et al., 2020; Geertsema et al., 2022; Lacroix et al., 2022); however, few such observations have been made in the coterminous United States. Despite the lack of observed climate-driven landslides in the coterminous U.S., these high elevation slope failures pose a potentially high risk to an ever-increasing number of people who spend time in alpine environments. RMNP was visited by ∼4.4 million people in 2021, making it the 14[th] most visited National Park in the US (NPS, 2022). As the climate warms, developing awareness around future alpine hazards within RMNP and across the broader western U.S. Cordillera is an important priority.

### 1.1 Stability of alpine topography

A warming climate has led to a warming of permafrost, glacier retreat, and growing instability in alpine regions across the globe (Patton et al., 2019; Shan et al., 2014). This instability is readily observed in global glacial mass inventories, with 69% of global glacial mass loss between 1991-2010 being attributable to anthropogenic climate warming and only 25% of mass loss being due to anthropogenic climate warming in the 1851-2010 period (Marzeion et al., 2014; Hugonnet et al., 2021). An increase in rock fall, landslides, and glacier retreat has been well documented in the European Alps and the alpine regions of Canada and Alaska(Cossart et al., 2008; Deline et al., 2021; Kos et al., 2016; Geertsema et al., 2022). For example, 70% of the rock falls to occur on the Mont Blanc massif since 1947 took place after 1991 with 83.5% of rock falls surveyed between 2003 and 2014 originating in areas modeled to have permafrost (Ravanel and Deline, 2011; Deline et al., 2021). Some of these destabilization events have received international news coverage (e.g. areas below the Planpincieux glacier in Italy on the flanks of Mont Blanc) and resulted in intermittent area closures and evacuations (Dematteis et al., 2021; Giordan et al., 2020). Other events have been more remote and garnered interest from the scientific community but have not had an impact on a broader population (e.g. Lipovsky et al., 2008).

So far, few notable increases in instabilities in the alpine regions of the conterminous United States have been reported. This is despite the observations of sporadic permafrost across states such as Colorado above 3200 m a.s.l. and discontinuous per-

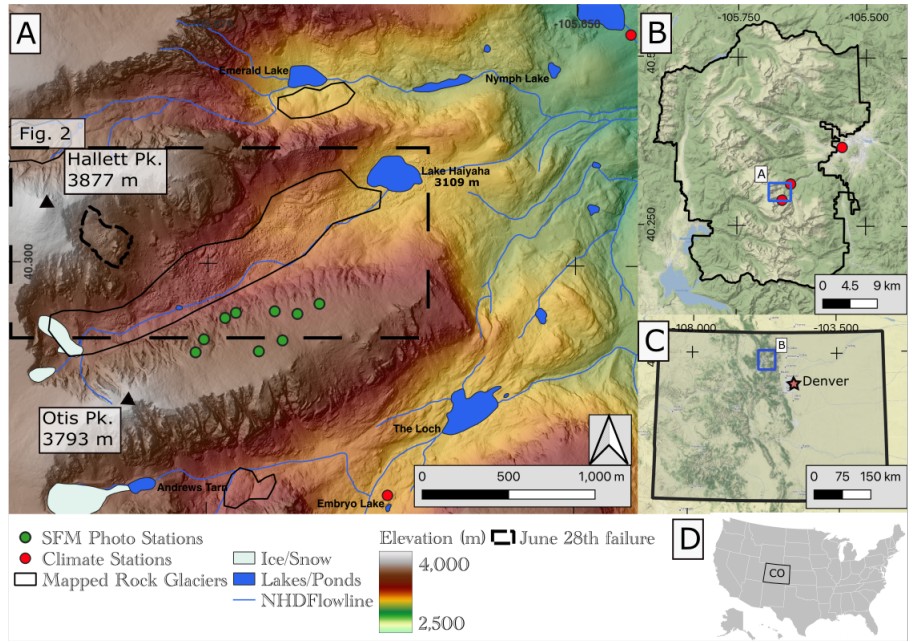

**Figure 1.** Overview of landslide area. **A)** Hillshade with hypsometric tint from 1 m lidar Digital Elevation Model (DEM) of the Chaos Canyon landslide and surrounding topography. Highlighted are the outlines of mapped rock glaciers (Johnson et al., 2021); the nine photo locations used in the structure from motion survey, and the two climate stations used in climate reconstructions for the slide. Lakes, streams, and ice/snow bodies are from the National Hydrography Dataset (NHD) (USGS, 2019). Imagery predates June 28th faillure. **B)** The location of panel **A** within Rocky Mountain Park. **C)** The location of panel **B** within Colorado, USA **D)** The location of Colorado (panel **C**) within the USA

.

mafrost above 3500 a.s.l., and the dynamic nature of mid-latitude permafrost which may quickly thaw due to climate warming (Ives and Fahey, 1971; Slater and Lawrence, 2013; Obu et al., 2019). It remains to be seen whether mountainous terrain in the conterminous US will experience a similar increase in events like those seen in Alaska and the European Alps (O'Connor and Costa, 1993) . The current geomorphological evolution of these alpine environments integrates both (1) the long-term response
to glacial retreat and glacial conditioning of the topography (i.e. on scales of 10s to 100s of thousands of years), and (2) the impact of a recent changing climate (i.e. on time scales of 10-100 years) via glacier retreat and permafrost degradation (Huggel et al., 2010; Stoffel and Huggel, 2012; Obu et al., 2019). Quantifying the dynamics of mountain landforms facing climate change is proving difficult, with the effects of recent decades of warming still yet to be seen in alpine landscapes (e.g., Christian et al., 2018). Predictions of the geomorphological response such as mountain slope instability to future climate scenarios
remain limited because the processes involved operate on interdependent timescales and are therefore difficult to characterize. Herein, we develop the tools necessary to analyze a high elevation, mid-latitude landslide and broaden our understanding of the stability of alpine slopes within a warming climate regime through this case study. We take a multidisciplinary approach by

characterising the relation between permafrost, topographic and climate forcings, and slope instabilities as they pertain to the June 28th, 2022 failure of the Chaos Canyon landslide.

## 1.2 The Chaos Canyon landslide

The Chaos Canyon landslide sits above treeline in the alpine reaches of Rocky Mountain National Park (Fig. A1). The pre-collapse landslide extended from ∼3450 m to ∼3660 m in elevation. Based on pre-collapse satellite imagery, the landslide is a diamicton, composed of grains ranging from fine sediments to large boulders (∼10s of m). The slide occurred along the contact between the Middle Proterozoic Silver Plume Granite and the early Proterozoic biotite schists (Fig. 2) with a moderate

foliation dipping to the southeast and toward Chaos Canyon (Braddock and Cole, 1990). A deposit in the bottom of Chaos Canyon was mapped by Braddock and Cole (1990); Johnson et al. (2021) as a potentially active rock glacier, with the landform that failed on June 28th mapped as Quaternary talus (Figs. 1, 2).

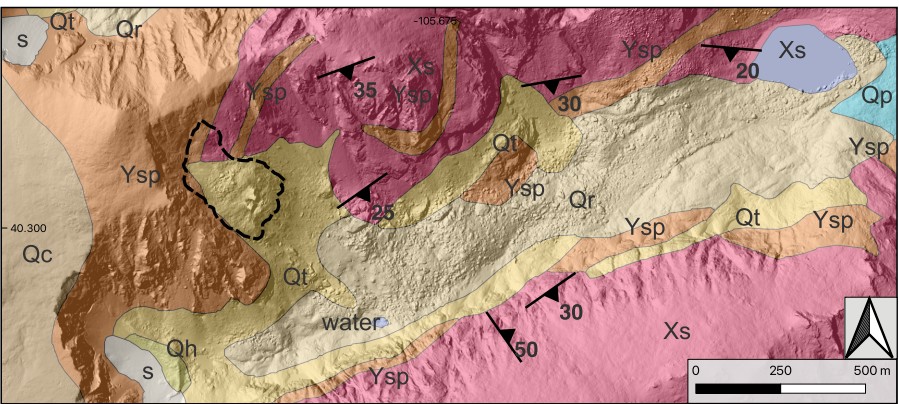

**Figure 2.** Geologic context of the Chaos Canyon landslide, perched above the canyon floor atop a contact between the Silver Plume Granite (Ysp) and a Biotite schist (Xs). Note there is a foliation dip toward Chaos Canyon at 35°. Other lithic designators are: Qc - Quaternary colluvium; Qr - Quaternary rock glacier; Qt - Quaternary talus; Qh - Quaternary till; s - snow; water bodies are not colored. Dashed black line is extent of June 28th, 2022 failure. Underlying geologic data is from (Braddock and Cole, 1990)

## 1.3 Primary Questions

This collapse spurred a series of questions at the scale of the landslide itself which also broaden to the scope of alpine landforms

across the conterminous United States:

1. Was the landslide moving prior to its collapse on June 28th ?

2. What were the climatic trends leading into the collapse?

3. Can we constrain the volume of the slide? What is this volume?

4. Can we ascertain the presence or absence of permafrost? Is there evidence of degradation of permafrost through time?

5. Can we use slope stability modeling to further evaluate the rheology of the slide and gain insights into the processes at work? For instance, can we quantify the role of groundwater?

6. What does this work tell us about the other landforms and their stability across the park and western Cordillera which is rapidly warming?

## 2 Methods

To answer questions posed above, we undertook a multidisciplinary investigation of the Chaos Canyon landslide and the events that led up to the failure. We combined both satellite-based image correlation using optical data and interferometric synthetic aperture radar (InSAR) to constrain movement of the landslide prior to June 28th, 2022. A climatological analysis examined the role of snowmelt and whether the timing of the collapse coincided with peak snowmelt. To assess the landscape evolution impacts of the landslide, we created a Structure from Motion (SfM) model. We further developed climatological analysis by

modeling the potential presence of permafrost or interstitial ice within the landform. Finally, we conducted a slope stability analysis to evaluate the potential factors at play in the stability of the deposit prior to collapse.

### 2.1 Remote sensing

To understand the processes at work leading up to and during the June 28th collapse in Chaos Canyon, it is important to investigate characteristics of landslide motion prior to collapse. As in-situ survey data are unavailable, we utilized two complementary

remote sensing techniques: InSAR and image correlation.

#### 2.1.1 InSAR

InSAR is a remote sensing technique that can be used to measure millimeter-scale displacement of the ground surface from space (Bürgmann et al., 2000). Interferograms are InSAR-derived maps containing information about surface displacement between two acquisition times along the satellite line-of-sight (LOS). To investigate rates of displacement of the landslide

prior to the collapse on June 28th, we created interferograms using data acquired by the Copernicus Sentinel-1 A/B satellites. Specifically, we processed and analyzed all possible short temporal-baseline ($\leq$24 days) Sentinel-1 interferograms overlapping the study area from July 15th-September 15 of 2015-2021 using the Jet Propulsion Laboratory InSAR Scientific Computing Environment version 2 software (Rosen et al., 2012). Short temporal baselines were chosen to limit possible unwrapping errors during interferogram processing, which are frequently caused by features moving more than half the radar wavelength

($\sim$2.8 cm for Sentinel-1) between acquisitions. Stacks of 34, 35, and 42 interferograms were derived from ascending (satellite flight north and looking east) track 78, ascending track 151, and descending (satellite flying south and looking west) track 56, respectively. Interferograms were processed with 6 looks in range and 1 look in azimuth, resulting in roughly 14 by 14 m pixel spacing. A 2017 U.S. Geological Survey (USGS) 3DEP Digital Elevation Model (DEM) with 10 m pixel spacing was used to

remove the topographic component of the phase and geocode the interferograms. For each of our three interferogram stacks, we removed low quality and noisy interferograms using a coherence threshold >0.6. Coherence is related to the similarity of scatterers in the images that form interferograms; low coherence indicates that the target surface is changing appreciably and displacement signals are unreliable. We lastly computed the pixel-wise median where coherence was >0.4, yielding a single median velocity map for each track (e.g., Fig. 4).

### 2.1.2 Image correlation

We used image correlation to measure 2D ground displacements (east-west and north-south) at the Chaos Canyon landslide based on Google Earth and PlanetScope images. We examined all available historical Google Earth images of the site and identified two high-resolution photos that had little snow coverage and no visible artifacts. The images selected were from 9/2016 and 8/2019. Before exporting the images, we turned off terrain and 3D effects, as well as image compression and filtering. We selected the maximum available image output resolution of 8192 x 4925 pixels from Google Earth Pro. Scaling of the images from pixels to meters had to be manually evaluated; for this we measured several distances on each image in Google Earth and then determined the corresponding pixel dimensions, resulting in a scaling value of 0.21 m/pixel. We then applied a Fast Fourier transform-based image correlation approach that first aligns the image pair with a co-registration routine, then evaluates internal misalignments using a moving window to measure displacements in the plane of image (Bickel et al., 2018). We used a window size of 256 x 256 pixels with 50% overlap, resulting in ∼27 m resolution outputs, and a vector based post-processing filter (for details see Bickel et al., 2018).

In addition, we calculated the time-dependent displacement of the landslide between 2017 and 2021 using PlanetScope imagery (3 m pixel resolution). We selected 5 images acquired during the snow-free period (either from August or September each year). Unfortunately, there were no snow-free images in 2022 prior to the failure of the slope. The PlanetScope images are orthorectified and have radiometric, geometric, and sensor corrections applied (Planet-Team, 2017). We performed image correlation on 10 image pairs with a minimum time span of ∼1 yr and a maximum time span of ∼4 yrs between images (Table A1). For image correlation analysis using PlanetScope imagery, we used the Outlier-Resistant Correlator (OR-Corr) subpixel image correlation method (Milliner and Donnellan, 2020). We used a 33 x 33 pixel correlation window with a step size of 9 pixels, resulting in 27 m pixel resolution displacement maps. We then used the MintPy timeseries software (Yunjun et al., 2019) to invert for the time-dependent motion of the landslide.

## 2.2 Climate analysis

To better understand the climatic circumstances of the June 28th collapse, we analyzed the records at the Bear Lake SNOw-pack TELemetry Network (SNOTEL), from the United States Department of Agriculture and Natural Resources Conservation Service, 2022) located ∼3 km to the northeast of the landslide. Specifically, we wanted to test the hypothesis that snowmelt may have contributed to the catastrophic failure. We extracted the Bear Lake Climate SNOTEL (Fig. 1) temperature record going back to 1991 (NRCS, 2023). This meteorological site is located at 2903 m above sea level (asl). We made use of the Global Historical Climatology Network site (USR0000CEST) located in Estes Park at 2382 m asl (Fig. 1) to determine a local

environmental lapse rate of $4.65 \times 10^{-3}$ °C m$^{-1}$. This lapse rate was calculated by examining the average daily temperature difference between these two sites and dividing by their difference in elevation. We then shifted the temperature record collected at Bear Lake to reflect the environment at the top of the slide at ~3147 m asl. We made a cumulative positive degree day sum (PDDS) of the temperature record representative of conditions at the top of the slide (Braithwaite and Hughes, 2022). PDDS is a cumulative sum of the average daily temperatures greater than 0 °C. By applying a global average snow ablation rate of ~4.5 mm day$^{-1}$ °C$^{-1}$ (Anderson et al., 2014), we estimated spring snowmelt atop the Chaos Canyon landslide.

## 2.3 Structure From Motion

A DEM of the June 28th collapse, was obtained using SfM and terrestrial photogrammetry. Data collection took place on July 8, 2022, ten days after the collapse. Photographs were taken from 9 different stations on the East of Otis Peak ridge providing a direct view of the landslide (Fig. 1; Table A2). Photo acquisition was made between 10:30 and 15:00 ensuring ideal light conditions. We used a Sony Alpha a7S III Mirrorless Digital Model with a Tamron 28-200mm f/2.8-5.6 Di III RXD Lens mounted on a tripod. The following settings were used for all pictures: a focal length of 8.0, iso 400, zoom of 70 mm and shutter time between 1/800 and 1/1250 s. Pictures were taken with approximately 80% of overlap between each other. The coordinate of each station was recorded using a handheld GPS (Garmin GPSmap 64s). This post-collapse DEM was compared to a reference DEM (before the event) to (i) record the new geometry of the area and (ii) quantify the amount of erosion and deposition involved. The reference 1 m DEM is provided by the USGS program National Map 3DEP and was acquired in 2017.

The 3D point cloud was created with Agisoft Metashape Professional using 305 photographs. The photographs were first aligned with high accuracy, setting the key point limit to 4000 and the tie point limit to 100,000. The resulting sparse point cloud was filtered using a reconstruction uncertainty criterion of 300. The camera locations were estimated with a total error of 40 m (X error of 22.1 m, Y error of 10.3 m and Z error of 31.7 m). We used the reference 1-meter DEM from USGS to create virtual Ground Control Points (GCP; Fig. 3). Those GCPs were chosen outside and around the landform affected by the destabilization, on bedrock features that are recognizable on both the reference DEM and the photographs acquired after the event. This approach produced 5 GCPs with total location error of about 0.7 m (X error of 0.04 m, Y error of 0.09 cm and Z error of 0.13 cm, see Table A3). Finally, we produced a dense cloud made of $20 \times 10^6$ points that we converted into a DEM of 0.26 m/pixel resolution with a point density of 14 points/m$^2$. Irregularities in the obtained DEM were further removed using the following sequence of steps. First, an iterative procedure was used to identify all local depressions in the DEM (Barnes et al., 2014). These were subsequently filled using an Inverse Distance Weighing algorithm. This step was repeated until all local minima were removed. In a second step, positive spikes in the DEM were identified using a slope-based DEM-filter. These spikes were subsequently removed and resulting gaps were interpolated using the iterative procedure described above. Then the smoothed post-collapse DEM was subtracted from the pre-collapse DEM (2017) to construct a DEM of difference (DoD), from which volumes of erosion and deposition can be calculated. We also calculated simple empirical length and height metrics to compare this landslide to other landslide inventories.

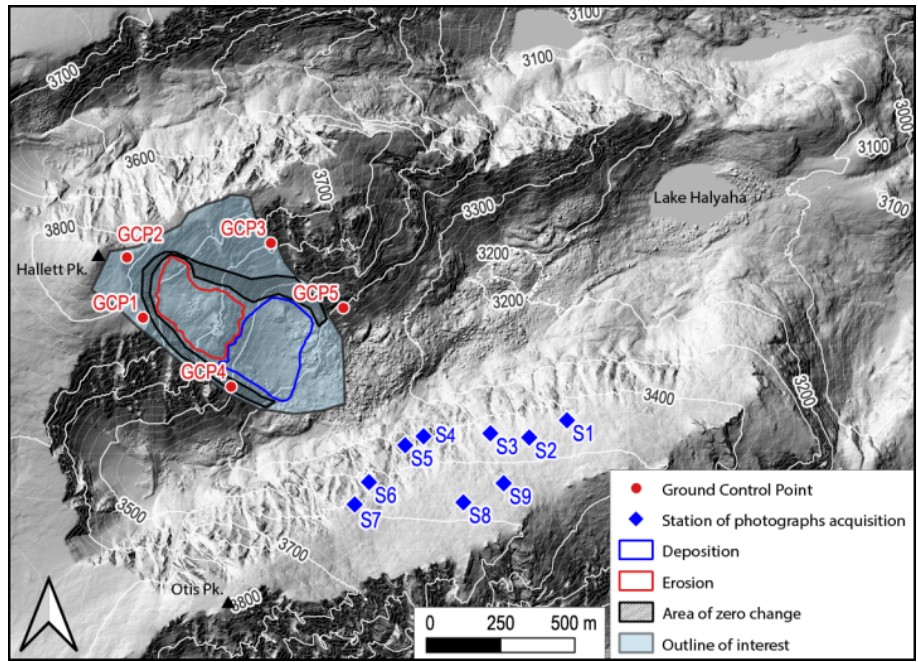

**Figure 3.** Location of the stations where the photographs where acquired (S) and the ground control points (GCP) used to create the post-collapse DEM. Hillshade and elevation lines were computed using the 1-meter DEM provided by the USGS program National Map 3DEP and acquired in 2017. Areas of erosion and deposition were derived from differencing the 2017 DEM and our July 8th SfM DEM

## 2.4 Permafrost modeling

To explore the soil and bedrock temperature profile at the time of the June 28th collapse, we used a coupled model of snow and permafrost, consisting of an empirical snow model (ECsimplesnow) and the Geophysical Institute Permafrost Laboratory (GIPL) (Brown et al., 2003; Jafarov et al., 2012; Overeem et al., 2018). GIPL is a one-dimensional heat flow model, simulating ground temperature evolution and the depth of the active layer by solving non-linear heat equations with phase change. GIPL is often set up as a spatial grid consisting of adjacent columns; however, we have little information on the soil and debris cover thickness or spatial variability in snow depth. Therefore, we chose to model the annual evolution of subsurface temperature for a single vertical column.

We initialized subsurface properties needed for the GIPL model with soil characteristics from global soil data released in SoilGrid (Hengl et al., 2014). Key properties include soil texture and water content. We had no in-situ data, but the pre-collapse satellite imagery indicated an existing diamicton – consisting of sediment and boulders. For our simulations, we assumed ∼30 cm regolith soil, 2.7 m debris and coarse, un-weathered sediment, and bedrock beyond that depth. Coarse-grained sediment and bedrock then define the frozen and unfrozen thermal conductivity and heat capacity according to relationships established from laboratory experiments (Goodrich, 1982; Kersten, 1949; Schaefer and Jafarov, 2016).

We ran the combined model over the 2021 water year (October 1,2020 to September 30, 2021), with a time-series of daily air temperature and precipitation, as extrapolated from observations at the Bear Lake station (as described in Section 2.2). Soil or bedrock temperatures were strongly modulated by snow cover over winter, due to its low thermal conductivity (Zhang, 2005). The effect is complex, and ground temperature can either be lower or higher than the snow surface or air temperature, depending upon the timing, duration, and thickness of the seasonal snow cover and the air temperature history. The spatially variable but locally large wind-drifted snow accumulations in cirques in the Rocky Mountains act as a thermal insulator on an annual basis. Our regional snow model, from original empirical parameterization of Brown et al. (2003), used daily precipitation input over the water year and combined this with a snow classification map (Sturm et al., 1995) to establish snow thickness and density. This is numerically implemented in the ECsimplesnow model and coupled with the GIPL model. The landslide sits between∼3450 m and ∼3660 m a.s.l., which is well above the regional tree line, so vegetation coverage was set to be 'open terrain', meaning that trees or extensive shrubs do not impact the snow properties.

## 2.5   Slope stability modeling

To explore the conditions that led to collapse of the deposit, we used the limit equilibrium analysis (e.g., Duncan, 1996) program Slide2 (Rocscience, 2021). For the analyses, we imported pre-collapse topography based on the LiDAR DEM of Rocky Mountain National Park. To estimate the boundary between the pre-collapse deposit and the underlying bedrock we extrapolated a surface under the pre-collapse deposit based on known bedrock outcrops on either side of the post-collapse deposit in QGIS. The underlying bedrock topography was estimated by adjusting contours, converting the contours to point data, and then re-rasterizing the data using the *GDAL_rasterize* command in QGIS.

Slide2 was used for the limit equilibrium analysis. Because the collapse occurred during the snow melt season, we explored the stability of the pre-collapse deposit to changes in water table depth. We chose to focus on this parameter related to slope stability because such little information was available regarding the landslide before its June 28th collapse. Most of the information we do have is on the rate of snowmelt and presence of permafrost. Material properties of the landslide were tuned based on our knowledge of the site from in-situ observations, pre-collapse velocity data, and permafrost modeling. We modulated the landslide and underlying bedrock densities – (2000 kg m$^{-3}$ for the deposit and 2700 kg m$^{-3}$ for the bedrock). We added a shear plane between the deposit and bedrock. We then tuned the properties of the shear plane and deposit to represent a global minimum factor of safety slightly greater than 1 (actual FoS of 1.037) to represent the observed condition that the landslide was moving prior to collapse (see 3.1.2); we hypothesized this indicates a well developed shear plane developed (Fig. 11). For the shear plane, we assume a cohesion of 1 kPa and a friction angle of 30°, while for the ice-cemented deposit we assume cohesion of 250 kPa and a friction angle of 50°.

In this model, we assume a Mohr-Coulomb failure criteria for the pre-collapse deposit and underlying bedrock (Labuz and Zang, 2012). We compared the pre-collapse global minimum factor of safety in cases with and without a slight rise in local water table. The factor of safety is the ratio of resisting forces and driving forces (e.g., Duncan, 2000). When the factor of safety is greater than 1 the slide is stable; when the factor of safety is less than one the slide will fail. The specific details of the model domain are described below.

## 3 Results

### 3.1 Pre-collapse movement

#### 3.1.1 InSAR

For all tracks, median coherence within the landslide boundaries was low ($0.36 \pm 0.17$ for ascending track 78, $0.37 \pm 0.16$ for ascending track 151, and $0.52 \pm 0.16$ for descending track 56, Fig. 4D). Twelve wrapped interferograms from ascending tracks 78 and 151 show clear evidence of landslide displacement beginning in August 2015 (Figs. 4B, A2A, A3). Median LOS velocities in the landslide were not distributed in a manner consistent with cohesive downslope displacement. Instead, we observed patches of apparent upslope and downslope LOS velocity, along with patches of apparently stable area, are scattered across the landslide surface in no clear pattern, which is due to unwrapping errors caused by the high deformation gradient (Figs. 4B, A2A, 6; Itoh, 1982; Handwerger et al., 2015). Thus, the median velocity does not provide a reliable indicator of landslide activity. By comparison, for all tracks, a rock glacier in the cirque to the north has a more spatially consistent downslope velocity signal (Fig. 4B), while rocky, flat areas above and below the landslide appear mostly stable outside of topography-correlated atmospheric noise.

#### 3.1.2 Image Correlation

#### 3.1.3 Google Earth

Results of Digital Image Correlation (DIC) analysis are shown in Figure 5. We observe a broad area of relatively large-magnitude displacement corresponding to the top surface of the landslide: in the $\sim$3 years (i.e. the period 2016 - 2019) between images this area moved on average $10.5 \pm 0.5$ m (Fig. 5B). Maximum displacements of 11.5 m were found on the northern portion of the slide surface. The main landslide body showed southeast trending movement with consistent displacement vectors (Fig. 5C). Areas of low correlation between images, and those which were filtered in post-processing, were located in the region covered by snow in the 2019 image (Fig. 5A) and near the toe of the slope (Fig. 5B). We also observed an adjacent movement near the northern head of the landslide with lower magnitude displacements ($\sim$2 m) and more southerly oriented movement than the main body. Here it appears that a portion of the talus adjoining the main slide body is moving in response to motion of the main slide away from its toe. A visible scarp had developed in the 2019 image at the head of this smaller sliding body. The mean displacement azimuth for the main slide body is 117 degrees clockwise from north, while the smaller northern portion is moving at an azimuth of 162 degrees. Finally, movement detected on and at the toe of the steep frontal slope has similar orientations only slightly lower magnitude than at the crest ($\sim$8 m). This may indicate evidence supporting a basal sliding mechanism for slide movement, that together with some amount of internal shear could generate the displacement pattern measured (Fig. 5B).

We can assess uncertainty in our image correlation results by measuring estimated movements in stable areas not anticipated to experience movement in the period between image acquisitions. For this, we evaluate a portion of the results on the broad

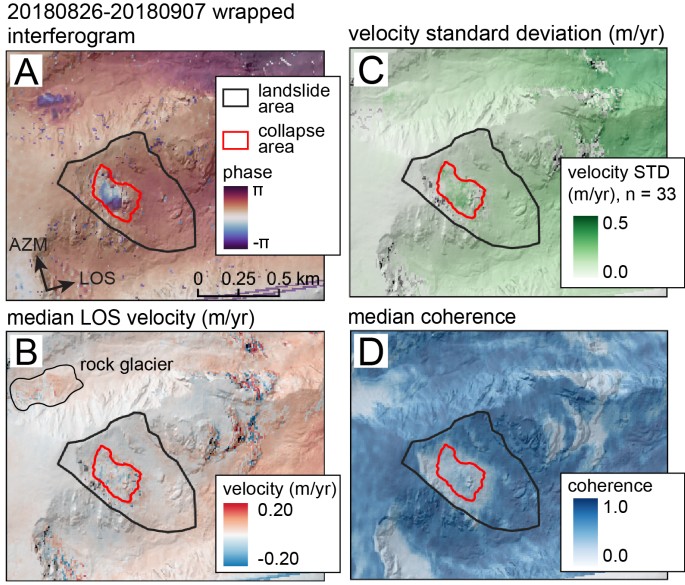

**Figure 4.** InSAR-derived LOS velocity of the landslide prior to the June 28th collapse. Interferograms come from Sentinel-1 ascending track 151. **A)** A wrapped interferogram from the summer of 2018 showing clear landslide deformation of roughly 10 cm in 12 days. **B)** Median LOS velocity of the failure area and landslide area. Interferograms with short temporal baselines and high median coherence on the landslide slope were used to create the velocity map. Positive values (red) correspond to motion away from the satellite along the satellite LOS. Note the spatial inconsistency of signals within the landslide. Negative velocity values at lower elevations are caused by topography-correlated atmospheric noise. **C)** Standard deviation of LOS velocity. **D)** Median coherence of the failure area and landslide area.

south slopes of Hallet Peak, and find that resolved mean movements are $0.3 \pm 0.1$ m, well below the measured displacement
of the landslide. While the image comparison shows significant displacement of the landslide between 2016 and 2019, no
information is available from this analysis on when this movement might have occurred during the interval. Finally, while the
relative displacements are robust, our use of Google Earth imagery requires empirical scaling assessment done on a case-by-
case basis and not benefiting from the pixel dimension information or metadata from the original acquisition. This leads to some
uncertainty in the absolute scaling of the displacement magnitudes. However, given consistent scaling measurements across
the images, together with independent measurements of displaced boulders identified visually in the images are consistent with
displacement magnitude inferred from image analysis.

### 3.1.4 PlanetScope

Image correlation of the PlanetScope imagery is in general agreement with the spatial extent and magnitude of the actively
deforming slope measured with the Google Earth imagery (Section 3.1.3). Our time series analysis reveals that the landslide
moved as a coherent unit but exhibited different rates spatially. The median cumulative displacement of the landslide was
$\sim$29 m $\pm$ 3.5 m ($\pm$ standard deviation) between 2017 and 2021. The maximum cumulative displacement was $\sim$39 m and

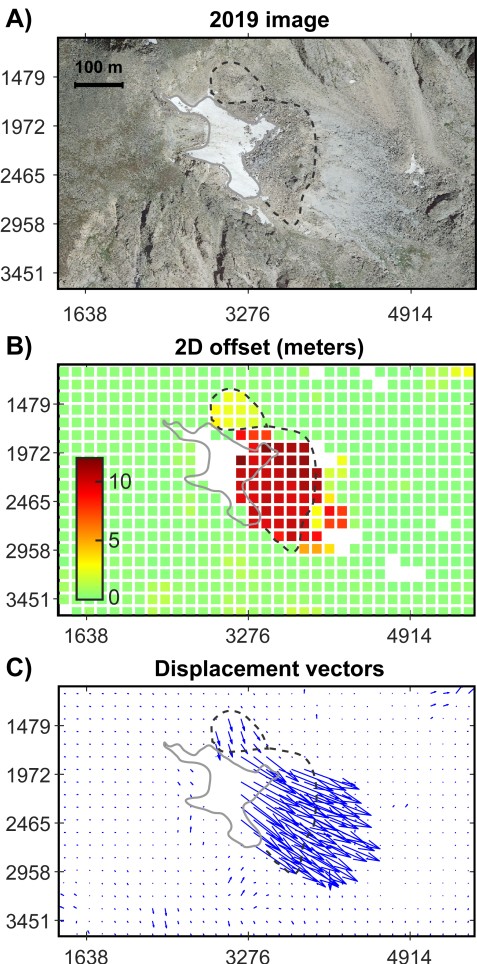

**Figure 5.** Results of digital image correlation for the Chaos Canyon landslide comparing Google Earth imagery from 9/2016 and 8/2019. **A)** 2019 photo of the slide with snow patch highlighted (grey solid line). Dashed lines highlight the crest of the main slide as well as a smaller adjoining slide thought to be secondary in response to undermining. **B)** Scaled 2D offset magnitude. Blank cells are areas omitted during filtering; these mostly occur in snow-covered areas and at the toe of the steep frontal slope. Movement of the main body is 10.5 m on average. **C)** Displacement vectors. The main Chaos Canyon slide and secondary adjoining slide show different displacement magnitude and orientation (dashed lines repeated from **A)**. Coordinates are pixels in east (x) and north (y) orientations; image scaling is approximately 0.21 m/pixel.

the minimum was ~20 m. The velocity increased monotonically between 2017 and 2021, but exhibited a distinct acceleration point starting in the summer of 2019. The median velocity across the landslide was <5 m/yr prior to 2019 and then increased to ~17 m/yr by 2021. This change in kinematics starting in the summer of 2019 suggests a change in stability conditions of the slope. We also assessed the uncertainty by examining the apparent movement in a stable area. We found that the stable slope exhibited apparent displacements < 1.3 m ± 0.27. We also explored the inverse-velocity relationship often used to predict

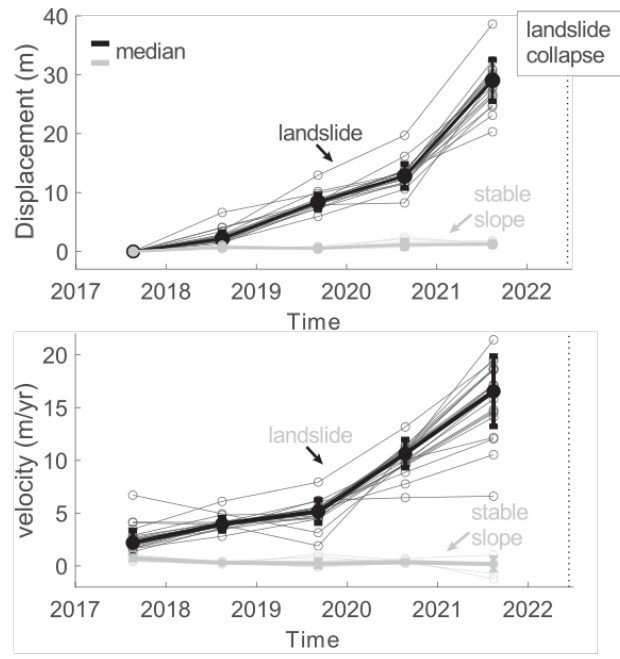

**Figure 6.** Image correlation results from Planet images taken in 2017, 2018, 2019, 2020, and 2021. **A)** The thin gray lines are horizontal displacements of every pixel mapped within the landslide. The black line is the mean displacement with 1-$\sigma$ bars. The thick gray line shows the same statistic but for stable areas outside the landslide. A greater sampling of Planet images reveals increasing displacements moving toward the present. **B)** The mean and 1-$\sigma$ uncertainty envelopes of velocity and for the landslide (black) and pixels examined outside of the footprint of the landslide (gray). The pixels outside the landslide show no systematic movement, as compared with the accelerating landslide.

landslide failure (e.g. (Fukozono, 1990; Voight, 1989), but found it yielded poor results. We did not pursue this line of inquiry further. For a direct comparison between our Google Earth and Planet imagery based methods please see Appendix Figure A5.

## 3.2 Snowmelt Rates

Our detailed climate analysis shows that the Chaos Canyon landslide collapsed as average daily temperatures were increasing to their summer peak ∼21 days later (Fig. 7A). 2022 was not atypical from previous years over the last three decades. It was not remarkably warmer than past years in the winter months, nor were spring temperatures significantly warmer than typical. However, the temperature series does indicate that the collapse may have taken place as warming increased the rate of snowmelt. This observation is further supported by the cumulative snowmelt we calculated at the elevation equivalent with

the top of the landslide (Fig. 7B). Snowmelt begins in April with melt increasing rapidly after June 1. Compared with the 1990-2021 seasons, 2022 had slightly higher melt than most of the previous 32 years, with the 11th highest calculated melt April - July 1. Notably, 2018, 2020, and 2021 had greater snowmelt than the 2022 season, making 2022 unremarkable in terms of snowmelt volumes. However, these calculations do indicate that the collapse took place during the peak snowmelt for the

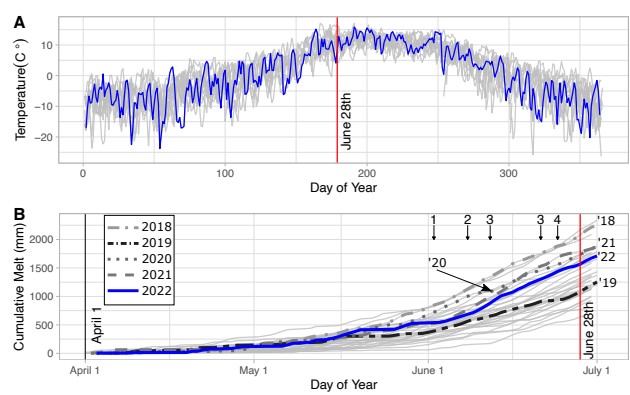

**Figure 7.** Climate analysis of the Chaos Canyon landslide. **A)** The mean daily temperature series estimated for ∼3668 m, the elevation at the top of the slide. 2022 is highlighted blue. 1990 thru 2021 temperature series are gray lines. The collapse occurred on June 28th, 2022. **B)** The calculated cumulative snowmelt for the past 32 springs. 2022 is the blue line with 1990-2021 snowmelt seasons shown in gray. Inset numbers highlight dates for frames from Planet Imagery in Figure 8. The snowmelt curves for the previous four years (2018, 2019, 2020, and 2021) are also highlighted.

spring season (Fig. 7B). This is bolstered by the Planet imagery in Figure 8. Panel A, June 2nd through June 24th, show a significant decrease in snow extent across the landslide in clear accordance with the snowmelt calculates in Figure 7B.

### 3.3 Change detection

The obtained difference map delineating areas of deposition and erosion associated with the collapse is highlighted in Figure 9. Delineation of these zones was done based on regions of zero surface change (green areas) between before and after event elevation data. Towards the edges of Figure 9 some artefacts start to appear in the data which can be attributed to lower point densities used during the SfM procedure in the boundary areas.

Uncertainty of the post-collapse DEM is obtained by calculating the Root Mean Squared Error (RMSE) between the reference LiDAR DEM and the post-event DEM for a region surrounding the erosion and deposition features that were minimally disturbed (visually determined in the field). The RMSE equals 2.39 m for this region, a value that is used to estimate uncertainty ranges on total erosion and deposition volumes. The erosion area covers 55,639 $m^2$ and experienced an average erosion of 24.09 $\pm$ 2.39 m or a corresponding volume of 1,340,000 $\pm$ 133,000 $m^3$. The deposition area covers 64,477 $m^2$ and experienced an average deposition of 19.52 $\pm$ 2.39 m or a corresponding volume of 1,258,000 $m^3$ $\pm$ 154,000 $m^3$. Erosion and deposition volumes are similar within uncertainty ranges. The deposited volume is slightly lower than the eroded volume which could be due to errors during the SfM DEM production or because of sediment evacuation towards downstream areas during and following the event.

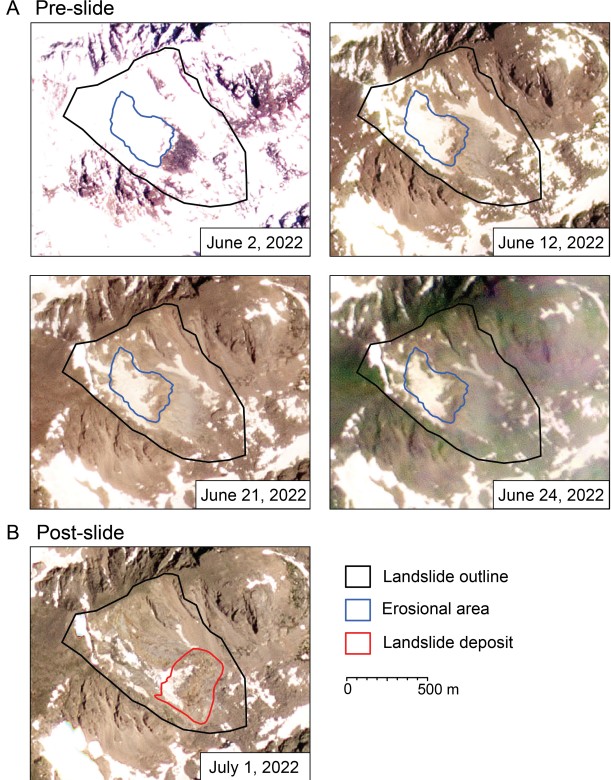

**Figure 8.** Planet imagery of upper Chaos Canyon in the days preceding the collapse. **A Pre-collapse)** Images from June 2nd, 12th, 21st, and 24th. Frames show decrease in snow coverage. Frames are also referenced in 7B. Satellite imagery further supports analysis that landslide collapse took place during a period of rapid snowmelt. **B Post-collapse)** The landslide deposit after the June 28th collapse is highlighted. Pieces of the permanent snowpatch have been translated with the landslide. Imagery thanks to Planet-Team (2017).

### 3.4 Permafrost modeling

The steep front of the pre-failure landslide averages an ∼40º slope, greater than a typical angle of repose could be indicative of interstitial ice holding the deposit together (Carson, 1977; Whalley and Martin, 1992). Further investigation of the potential for interstitial ice through ground temperature modeling indicates that permafrost conditions persist to at least 15 m depth in the pre-collapse deposit (Fig. 10). For the 2020-2021 model year minimum simulated surface temperatures was -4 °C, maximum simulated surface temperature was 16 °C, and the mean surface temperature was 2 °C. For transient simulations through the 2020-2021 water year the thaw front propagated to 0.96 m depth by June 28th, the date of failure. These results are only relevant for the portions of the pre-collapse deposit that were snow free. The deepest the simulated thaw front reaches, i.e. the maximum active layer depth, is ∼1.85 m at the end of the hydrological year in October (Figs. 10, A4). Importantly, these results do confirm the presence of continuous permafrost across the landslide but do support the presence of permafrost. Data on snow insulation toward the top of the slide, which would reduce the likelihood of permafrost stability is not available.

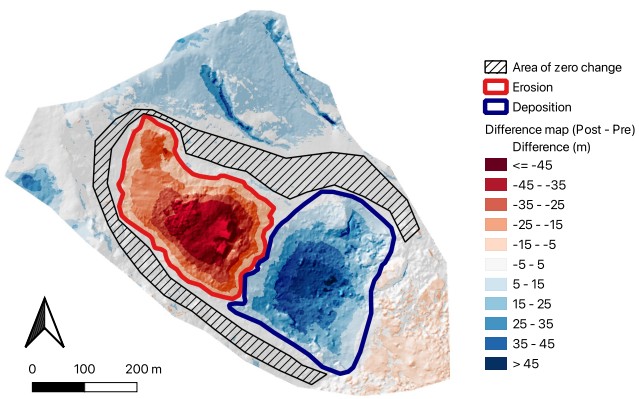

**Figure 9.** DEM of difference between Post and Pre-event topography. Negative values indicate erosion (red colors), positive values indicate deposition (blue colors), green values indicate regions of zero change. The dashed area indicates a regions where no visible change took place during the event and was used to calculate the accuracy of the DEM.

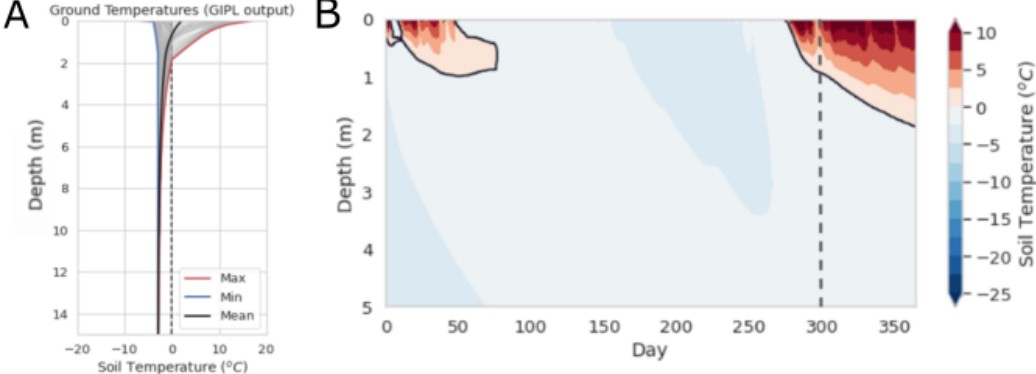

**Figure 10.** Simulation results of the combined snow and soil temperature model over the water year *Oct 2020-Oct 2021*. **Panel A.** Shows soil temperature as a function of depth, with even maximum temperatures below ∼2 m depth never reaching above 0°C. **Panel B.** Shows soil temperatures with depth at the Chaos Canyon landslide across a single water year. This result mirrors the depth of the active layer shown in **A.** The dashed vertical line is June 28th. Note the days are based on zero as October 1, the start to the water year.

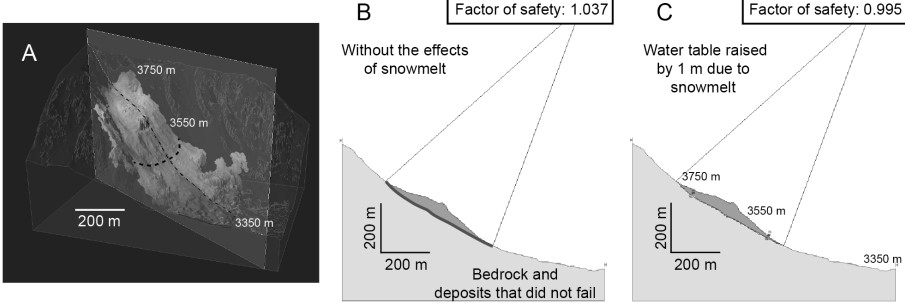

**Figure 11.** Output from limit equilibrium slope stability analysis. **A)** View of the pre-collapse topography in three-dimensions. The pre-collapse deposit is shown in gray. See appendix figure A6 for a color version. The thick dashed line represents the lower edge of the piece of the deposit that failed on June 28th, which is also shown in B and C. The thin dashed line shows the cross section also shown in B and C. **B)** Limit equilibrium analysis showing the global minimum slip surface and factor of safety for the simulated pre-slide deposit (purple) using the program Slide2. The failure surface (see the red line) is located at the transition between the purple body that moved during the collapse and the underlying material. **C)** Limit equilibrium analysis showing the global minimum slip surface and factor of safety for the simulated pre-collapse deposit with the effects of a water table 1 m above the contact between the pre-collapse deposit and the underlying bedrock.

## 3.5 Slope stability modeling

To explore the role of snow melt as a potential trigger of the landslide, we simulated a slight rise in the water table within the pre-collapse deposit. We expect that a basal shear plane is well developed at the base of the pre-collapse deposit, because the landslide has been accelerating and has undergone large displacements (Figs. 5,6). The inclusion of this water table reduced
the resistive forces in the system leading to a global minimum factor of safety of 0.995 and thus failure of the landslide. The pre-collapse deposit thus appears to be highly sensitive to a reduction of normal stress associated with a rising water table, likely associated with spring snow melt.

## 4 Discussion

Rapid environmental change is predicted to increase the occurrence of landslides and rock failures in high, alpine terrain
(Ravanel and Deline, 2011; Patton et al., 2019; Deline et al., 2021). Nevertheless, few alpine rock failures linked to climate change have been documented in conterminous United States. The Chaos Canyon landslide affords insights on the role of mass wasting in alpine terrain through abundant field and remotely sensed data as well as numerical scenario modeling. In the following, we discuss changes leading up to the event that could serve as tools to monitor future landscape stability.

## 4.1 Pre-collapse movement and potential causes of the June 28th collapse

The June 28th collapse of the Chaos Canyon landslide appears to have been an event lacking a well documented modern analogue in alpine regions of the conterminous United States. Image correlation results provide evidence of accelerating pre-collapse displacement at least as early as summer 2017, suggesting that unstable conditions developed over the course of years as opposed to within a single spring snowmelt season (Figs. 5, 6). It is important to note that inconsistencies in InSAR-derived median LOS velocities over the surface of the landslide suggest that our short-baseline interferograms suffered from
unwrapping errors. This interpretation is consistent with median LOS velocities of more than 0.93 cm/day during July 15th-September 15th of 2015-2021. Henceforth, we only discuss displacement rates derived from our image correlation efforts.

Initial rates of horizontal displacement were slow at ∼5 m/yr between 2017 and 2019, but the landslide rapidly accelerated after 2019 to a moderate rate of ∼17 m/yr (for velocity classes see: Cruden and Varnes, 1996). We identified no clear climatological forcing that led to the acceleration in 2019; however, there is the potential for a progressive weakening of the
failure surface beneath the landslide (Eberhardt et al., 2016) and/or potential slip localization to a single shear plane with landslide movement (Viesca and Rice, 2012; Scuderi et al., 2017). Both of these phenomena suggest through continued and repeated movement of the landslide, the landslide weakens and accelerates until its ultimate catastrophic failure. Similar behavior, wherein increases in sliding velocity result in weakening of the failure surface, has been modeled to occur in fault zones (Ito and Ikari, 2015). Given that the Chaos Canyon landslide has several years of slow and then accelerating movement, it is a
strong candidate for further analysis of potential rate weakening or shear zone development.

We can infer from the long-term trend in temperature the potential impacts of a warming climate on this alpine landslide. Temperature trends at the top of the landslide, approximated from the Bear Lake SNOTEL, show a clear warming signal across the past three decades (Fig. 12A, B), with a consistent positive temperature anomaly after 2009. Warming temperatures have been documented to correspond to landslides in both high elevation and high latitude landscapes, particularly where
permafrost or ground ice is present (e.g., Cossart et al., 2008; Deline et al., 2021; Patton et al., 2019). Increased temperatures lead to permafrost thaw and decrease slope stability through either a change in the physical conditions of the slope (e.g., reduced cohesion) or hydrologic conditions (e.g., increased pore pressure and hydrologic conductivity; Patton et al., 2019).

The presence of permafrost conditions (Fig. 10) in the pre-collapse deposit opens the possibility that ice occupied interstitial spaces between clasts (Kenner et al., 2017; Eriksen et al., 2018). If a continuous ice layer was present within the pre-collapse
deposit, snowmelt may have been channeled to locations where tension cracks were present through the pre-collapse deposit (e.g., Mutter and Phillips, 2012) . If abundant interstitial ice was present, then the entire pre-collapse body may have been moving down slope due to internal deformation and sliding. Satellite imagery of the steep front of the pre-slide deposit suggests that ice was exposed there in gullies; additionally, the pre-slide front was steeper than the angle of repose, indicating the presence of internal ice (Carson, 1977; Whalley and Martin, 1992). However, the lack of observed ice in the post-failure
deposit suggests that interstitial ice may not have been abundant, though internal heating during the slide could have also melted any internal ice (Pudasaini and Krautblatter, 2014). Further model simulations should explore the interaction between

percolating melt water and permafrost within the pre-slide deposit, to determine the potential for the presence of interstitial ice through the pre-slide body.

We hypothesize that permafrost within the slide deposit thawed and became intermittent due to the changing climatic conditions within Chaos Canyon over the last ~30 years, likely reducing the stability of the landform. Coincident with warming temperatures, we observed an increasing rate of horizontal displacement of the landslide over the past five years, with evidence suggesting both basal sliding and internal deformation (Figs. 5, 6). Thawing permafrost, combined with internal deformation and some surface cracking (Fig. 5), may have provided an increased availability of flowpaths for snowmelt and rain to penetrate into the slide mass, increasing the hydrostatic pressure and promoting destabilization of the deposit on June 28th (Figs. 7B, 11, 12C; Bogaard and Greco, 2016). Building on this discussion, we further posit that the June 28th collapse is likely due to a combination of factors: continued rate weakening and localization of a shear plain beneath the landslide in concert with a decrease in permafrost throughout the landslide. The former primed the landslide to be more sensitive to pore-water pressure increases, from even an unexceptional snowmelt year like 2022, and the latter provided more pathways for snowmelt to ultimately reach the failure plane and increase the porefluid pressures to point of failure. Additionally, the adverse dip of the underlying Proterozoic biotite schists may have played a role in mobilizing some of the landslide downslope (Fig. 2).

### 4.2 Landslide characterization

To better understand the processes at work in the Chaos Canyon landslide and to compare this landslide to other landslides, we used a few common empirical metrics that also inform landslide rheology and process. Landslide mobility is commonly expressed using the ratio between the maximum length of the sediment travel path (L) and the difference between the highest and lowest point impacted by the landslide (H) (Geertsema et al., 2009). The L/H ratio is a useful metric in hazard assessment because it indicates how far downstream landslide derived material can reach from any given source area (Iverson et al., 2015). Several studies have also shown a positive relation between the volume of mobilized material (V) and the mobility index (L/H). A compilation of this mobility index versus landslide volumes is given in Figure 13A. The Chaos landslide has a mobility value of L/H $\approx$ 1.8 and fits within ranges of earlier documented rockslides of similar volumes (Fig. 13A). The total inundated area of the Chaos landslide ($m^2$) also follows earlier documented trends between landslide extent and volume (Fig. 13B). An alternative mobility index for landslides is calculated as $A/V^{2/3}$. The mobility coefficient for the Chaos landslide of $A/V^{2/3} \approx 10$ is at the lower end of earlier documented mobility values of high-mobility landslides (Griswold and Iverson, 2008). Landslides with very high mobility coefficients are debris flows on ice or landslides in very wet environments where basal liquefaction plays a role (e.g. the OSO landslide where $A/V^{2/3} \approx 30$; Iverson et al., 2015). The lower value of $A/V^{2/3}$ for the Chaos landslide indicates a limited mobility where ice and water probably played a minor role in controlling landslide runout. However, this does not mean that the changes in internal ice and water were not potential contributory factors to the June 28th collapse. Perhaps this landslide would have been more mobile if it had collapsed prior to the observed warming we document (Fig. 12) which was likely paired with a decrease in interstitial ice.

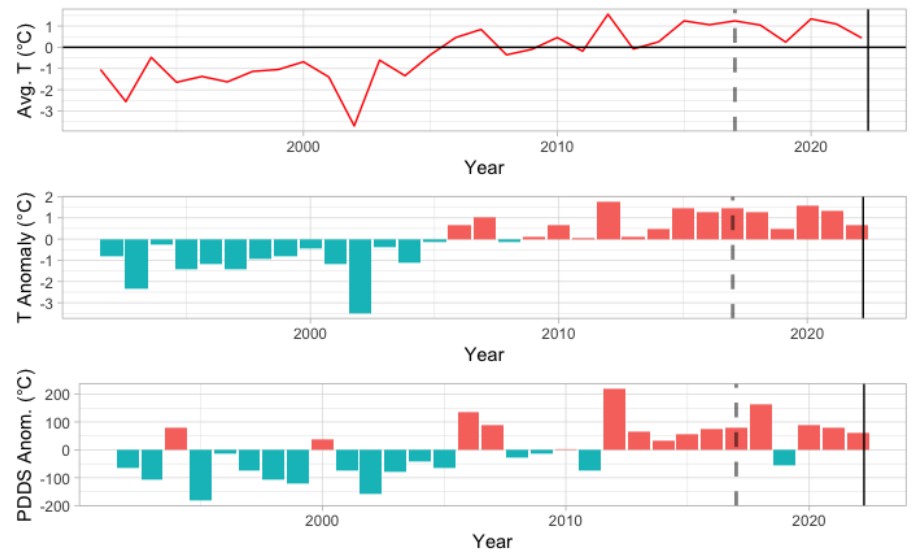

**Figure 12.** Temperature trends across the top of the Chaos Canyon landslide, using the Bear Lake SNOTEL record and the calculated local lapse rate. **Panel 1)** Average annual temperature from 1991 to the 2022 clearly shows a warming trend that begins in 2006. **Panel 2)** The annual temperature anomaly from the 31 year average, cooler colors correspond to a negative anomaly, warmer colors a positive anomaly. Here, the warming is even more visible as a positive temperature anomaly consistent across every year after 2009. **Panel 3)** The PDDS anomalies calculated for the first 6 months of the calendar year for the past 31 years at the slide elevation (Braithwaite and Hughes, 2022). While the PDDS anomalies show more fluctuation than the other indicators, after $\sim$ 2012 the PDDS is mostly greater than in the preceding 21 years. Moreover, PDDS has been directly linked with snowmelt and is a closer proxy to potential melt that could penetrate the slide mass. The gray dashed line is the first date of image correlation data, and the dark black line is the failure date of the Chaos Canyon slide.

### 4.3   Missing alpine landscape instabilities in mid-latitude North America?

The alpine regions of the coterminous United States have not, as of yet, seen a documented large increase in slope failures and rockfall linked to a warmer climate, with the the Chaos Canyon landslide documented herein a potential exception. While diagnosing this apparent lack of a landscape evolution signal tied to climate change is beyond the scope of this paper, we posit the following explanations: 1) The lack of direct observations (failures are happening but not being witnessed; e.g., Huggel et al., 2012). The coterminous United States is less densely populated than Europe so there are fewer opportunities for witnesses

to document failures in alpine regions. There has also not been a systematic inventory of InSAR data or permafrost across the region – put another away there are fewer people looking for instability. For instance, the Chaos Canyon landslide was actively moving meters per year in a National Park with millions of annual visitors, yet it went undetected until it failed in June 2022. 2) Much of this region of North America has yet to achieve a critical threshold in permafrost thaw, snow cover, annual average or extreme temperatures to permit such failures to become more common. Given current emissions and committed warming,

perhaps we will see more failures at an accelerating rate (e.g. Christian et al., 2018). Or 3) A null hypothesis. The types of

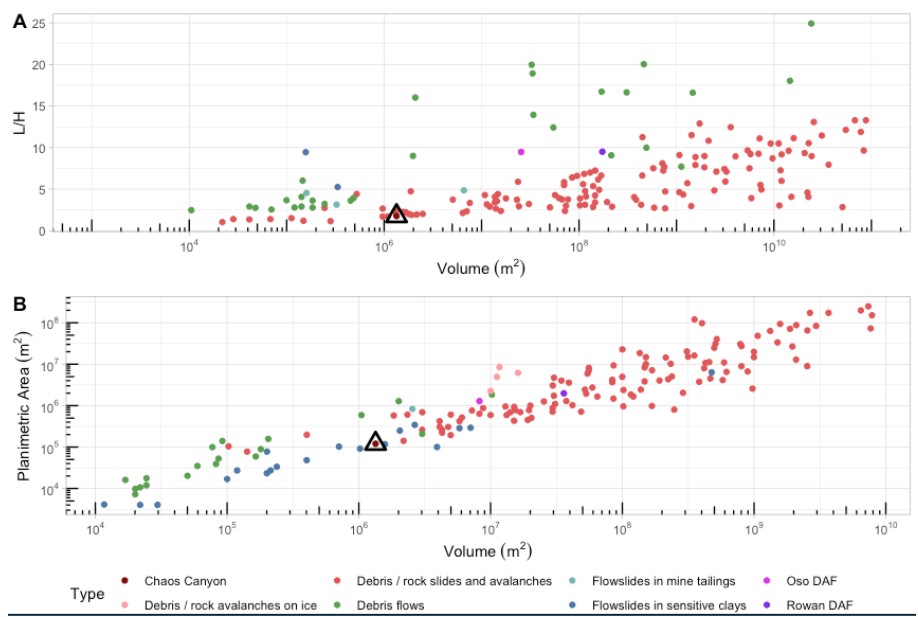

**Figure 13.** Landslide mobility indices. **Panel A.** The commonly used maximum landslide travel path (L) over landslide height (H) versus landslide volume. The Chaos Canyon landslide (marked as red dot with black triangle) has a low L/H ≈ 1.8. and falls within the field of other debris slides, rock slides, and rock avalanches (Iverson et al., 2015). **Panel B.** The Chaos Canyon landslide inundated an area consistent with power law relationship between landslide area and volume discussed in Griswold and Iverson (2008).

slope instabilities observed in the Alps, Canadian, Rockies, and Alaska are not occurring in the mid-latitude mountain ranges of the coterminous United States and may not occur. We encourage others to look to these hypotheses as an opportunity for more remote sensing and field explorations of the mountainous regions of the coterminous United States.

### 4.4 Ramifications for alpine landscape evolution

We are witnessing a transformative period in alpine landscapes (Patton et al., 2019). The past several decades have seen changes to the ice glaciers of the world (e.g. Kääb et al., 2018), the slopes adjacent to retreating glaciers (e.g. Dai et al., 2020), rock glaciers (e.g. Bodin et al., 2017; Eriksen et al., 2018; Marcer et al., 2019), alpine rockfall (e.g. Ravanel and Deline, 2011; Deline et al., 2021) and permafrost (Patton et al., 2019). We have shown evidence that the Chaos Canyon landslide falls within this spectrum of alpine landscape with instabilities likely tied to a warming climate (Fig. 12) and may represent an acceleration

of landscape evolution that has been continuing since the Little Ice Age ( 1650-1850 C.E. Benson et al., 2007). Our change detection methods reveal the slide translated ∼ 1,258,000 ± 150,000 m³ of material downslope. What we captured as part of this study is active landscape evolution, a process likely to be replicated in other alpine catchments.

The conditions documented in Chaos Canyon, an east-facing deglaciated valley which experiences higher rates of snow accumulation due to wind redistribution, are not unique in the Rocky Mountains. Where possible, mass movements should

be inventoried and monitored for changing displacement rates, such as those observed in Chaos Canyon, as an indicator of a potentially impending slope failure for safety monitoring and hazard mitigation. We particularly recommend these assessments in popular recreation areas throughout the mountain west, such as the National Parks. We demonstrate the utility of image correlation and the potential challenges of InSAR to detect mass movements, with repeat surveys serving to assess change over time. We caution other investigators to take a combined methods approach – if possible – pairing image correlation with InSAR to ensure rapidly deforming landforms are properly detected and their movement quantified with more than one method. Existing inventories indicate permafrost is sporadic at elevations above 3200 m a.s.l. and discontinuous above 3500 m a.s.l. in Colorado and the SW U.S. (Ives and Fahey, 1971; Obu et al., 2019). These continental scale studies could be improved with more regional inventories of permafrost. Locations where steep slopes, identified mass movements, and permafrost intersect need further monitoring to assess future hazards – particularly in areas with large numbers of visitors. There were luckily no injuries or casualties reported with the Chaos Canyon landslide, but the increasing popularity of hiking, climbing, and other alpine activities place more people in potentially dangerous locations. It is therefore imperative to understand how and where alpine slope instabilities may occur to minimize hazards in a warming world.

## 5 Conclusions

The June 28[th], 2022 Chaos Canyon landslide provides a unique opportunity to understand rapid landscape changes in the alpine environments of the Rocky Mountain west. Moreover, the event took place in the 14[th] most visited National Park in the country. Through our investigations, we have shown that the landslide, which translated $\sim 1,258,000 \pm 150,000$ m$^3$, was moving up to 17 m/yr in years prior to the June 28[th], 2022 failure, and was likely moving through a mix of both basal-sliding and internal deformation. These rates of translation were fast enough to cause unwrapping errors in our InSAR observations. With an L/H mobility metric of $\approx 1.8$, the Chaos Canyon landslide could be characterized as a debris slide, rockslide, or rock avalanche, meaning the landslide had limited mobility upon collapse. A comparable landslide with greater ice content would have higher mobility and potentially prove more hazardous. The slide occurred during the peak of spring snowmelt, and the preceding 13 years were particularly warm compared to the 31 year running average. Moreover, the first six months of the calendar year across the past 31 years have shown higher than average positive degree days, likely impacting the timing and rate of snowmelt. We hypothesize the several years of movement, potentially leading to rate weakening of the failure surface beneath the landslide combined with thawing of permafrost throughout the landslide body primed the landform for failure from an increase in pore-pressure. And as we documented through slope stability modeling, even a small increase in the water table leads to slope failure. We characterize the Chaos Canyon landslide as part of the broader alpine landscape evolution occurring across the high elevation and high latitude regions of the globe, and recommend the inventorying and monitoring of such alpine landscapes to better understand where these types of hazardous slope failures may be likely to occur under a warming climate.

*Code availability.* The code developed and associated with this publication is published on Zenodo. https://doi.org/10.5281/zenodo.7854068.

Other pre-existing and published tools are referenced in the text with sufficient detail for methods replication.

*Data availability.*

*Code and data availability.* The code developed and associated with this publication is published on Zenodo. https://doi.org/10.5281/zenodo.7854068.

*Sample availability.*

*Video supplement.* A short .mp4 video file of Planet images between 2017 and 2022 of the landslide is available in the Supplementary Material.

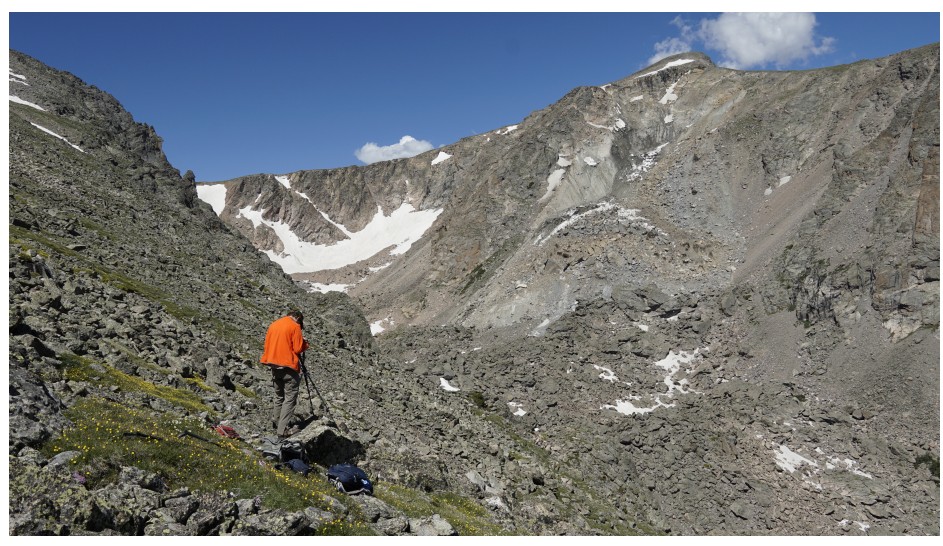

**Figure A1.** Photo of slide taken from south-southwest during SFM survey - one week after the collapse. Photo Credit: Benjamin Lehmann.

**Table A1.** Planet image pairs used in image correlation.

| Image Pairs | Start Date | End Date | Number of Days |
|---|---|---|---|
| 0 | 9/3/17 | 8/29/18 | 360 |
| 1 | 9/3/17 | 9/21/19 | 748 |
| 2 | 9/3/17 | 9/6/20 | 1099 |
| 3 | 9/3/17 | 8/28/21 | 1455 |
| 4 | 8/29/18 | 9/21/19 | 388 |
| 5 | 8/29/18 | 9/6/20 | 739 |
| 6 | 8/29/18 | 8/28/21 | 1095 |
| 7 | 9/21/19 | 9/6/20 | 351 |
| 8 | 9/21/19 | 8/28/21 | 707 |
| 9 | 9/6/20 | 8/28/21 | 356 |

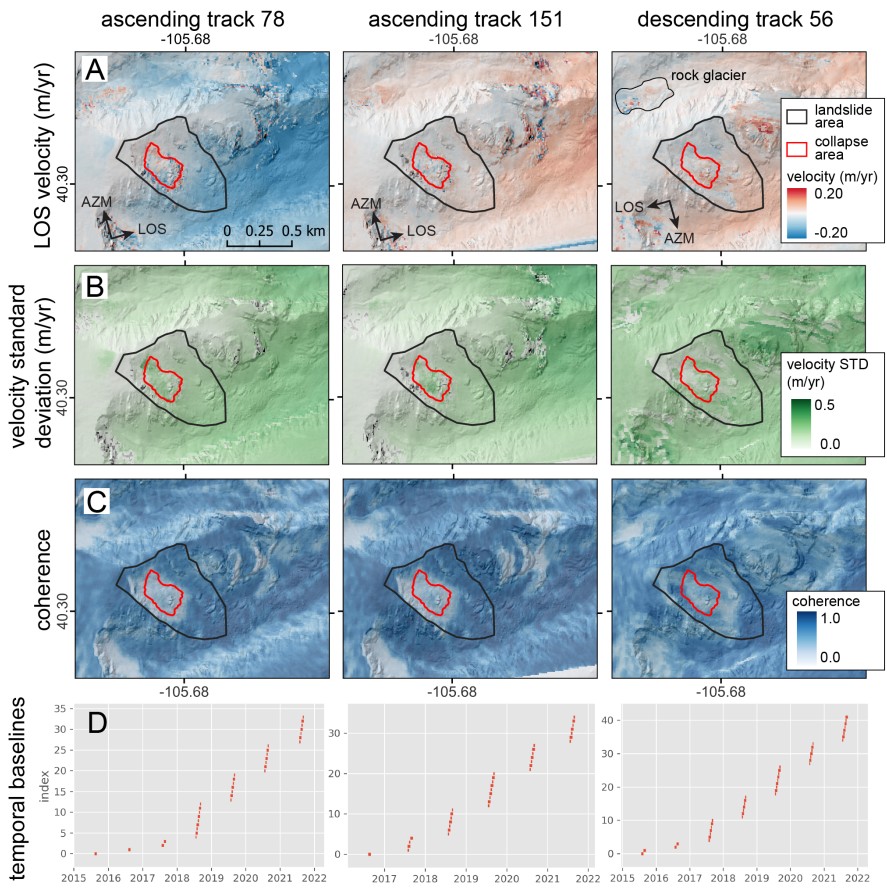

**Figure A2.** InSAR-derived median LOS velocity of the landslide feature prior to catastrophic failure. Sentinel-1 interferograms from ascending track 78, ascending track 151, and descending track 56 with short temporal baselines and high median coherence on the landslide slope were used to create each velocity map. **A)** Median LOS velocity of the landslide feature and surrounding area. Positive values (red) correspond to motion away from the satellite along the satellite LOS. Note the spatial inconsistency of signals within the landslide. Apparent topography-correlated displacements are caused by atmospheric noise. **B)** Standard deviation of median LOS velocity. **C)** Median coherence of the landslide feature and surrounding area. Coherence is related to the similarity of scatterers in the images that form our interferograms. Note the low coherence over the landslide feature, which indicates that the surface of the feature is changing appreciably. **D)** Temporal baselines of all available 6, 12, and 24-day Sentinel-1 interferograms from late summer. Each bar spans the temporal baseline of a single interferogram, beginning and ending at the primary and secondary acquisition dates.

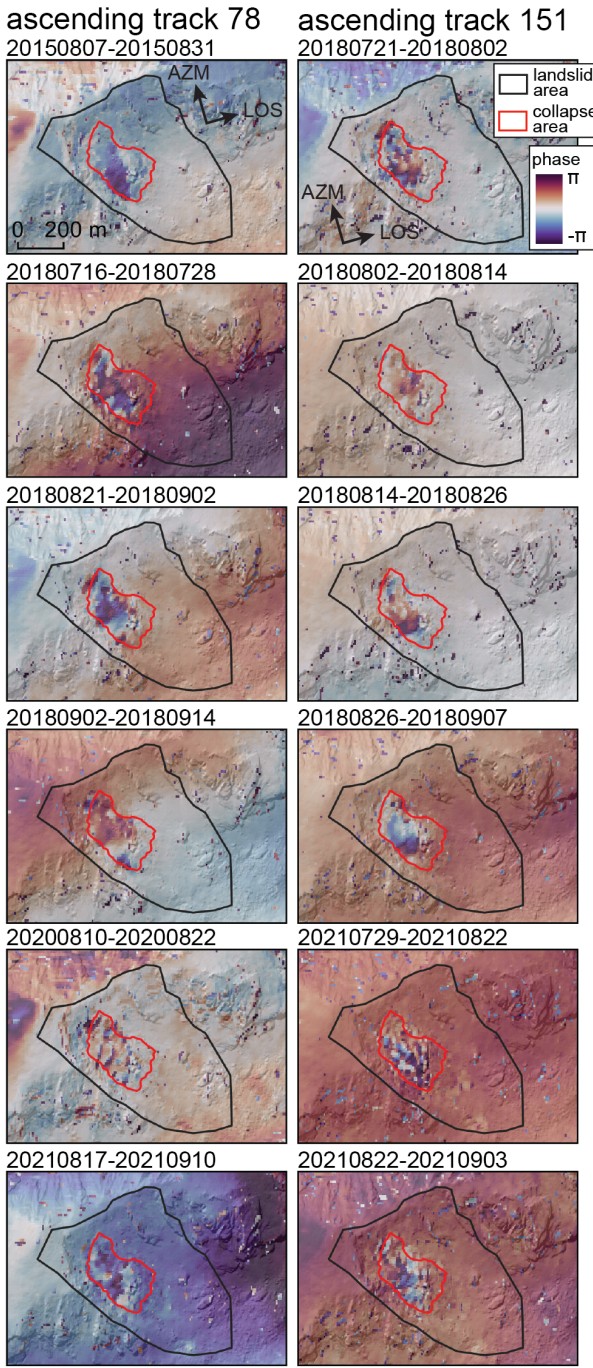

**Figure A3.** Wrapped short-baseline interferograms showing clear evidence of landslide displacement. Displacements of more than twice the radar wavelength, shown here, result in errors during unwrapping. Displacement signal is not as clear in other interferograms due to strong atmospheric noise and/or low coherence of the moving area. (left) Interferograms from ascending track 78. (right) Interferograms from ascending track 151.

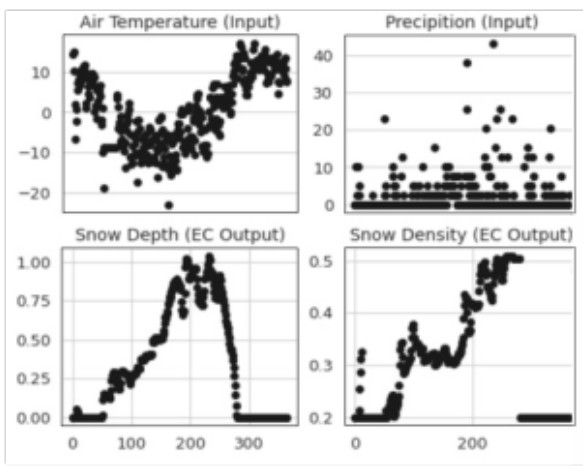

**Figure A4.** Simulation results of the combined snow and soil temperature model over the water year *Oct 2019-Oct 2020*. Daily air temperature and precipitation inputs are derived from correcting/extrapolating the lower elevation meteorological station at Bear Lake, RMNP. Snow depth and density are modeled by the snow model and are subsequent input for the GIPL heat conduction model.

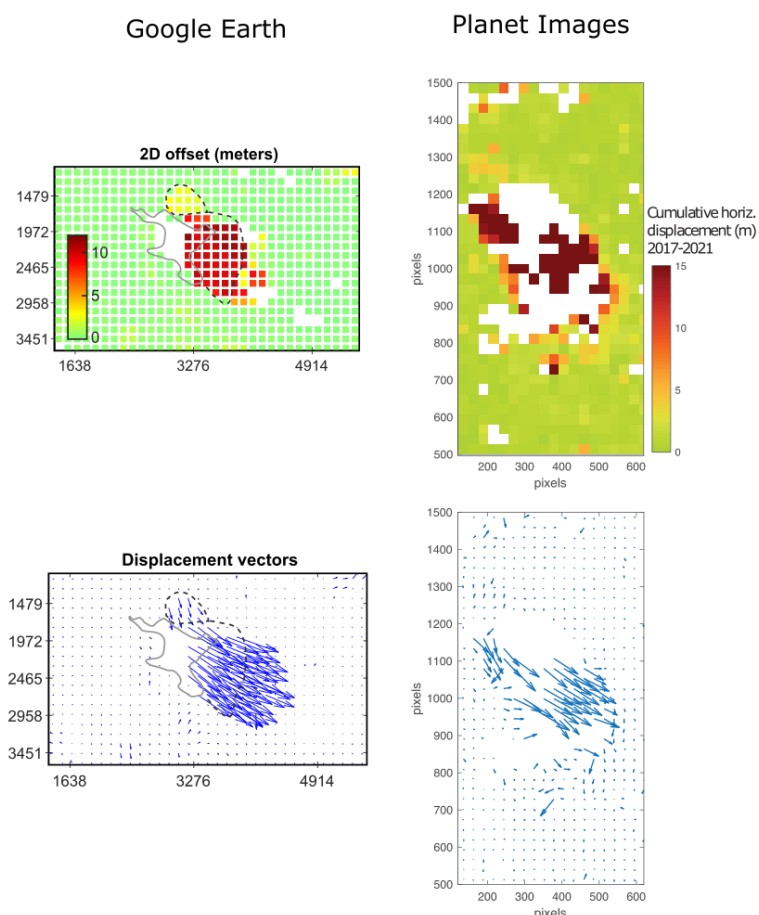

**Figure A5.** Direct comparison between pixel tracking methods. Note that the Google Earth results are the result from two years of comparison; the Planet imagery is multiple years of imagery.**Left panel** shows the Google Earth method, tracking horizontal displacements over the Chaos Canyon landslide. **Right Panel** shows the Planet imagery based pixel tracking results. While there are some differences between the results given the different frames used and different snowpack positions overall the results are quite similar, with the Planet images providing more frames across multiple years than the Google Earth images.

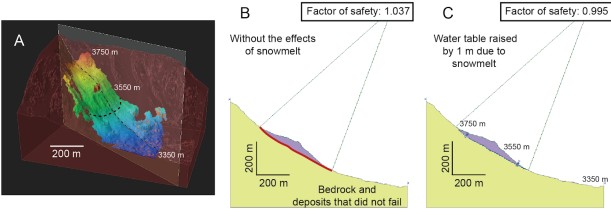

**Figure A6.** Output from limit equilibrium slope stability analysis. **A)** View of the pre-collapse topography in three-dimensions. The pre-collapse deposit is shown in red, yellow, and blue. The thick dashed line represents the lower edge of the deposit that failed which is also shown in B and C. The thin dashed line shows the cross section also shown in B and C. **B)** Limit equilibrium analysis showing the global minimum slip surface and factor of safety for the simulated pre-slide deposit (purple) using the program Slide2. The failure surface (see the red line) is located at the transition between the purple body that moved during the collapse and the underlying material. **C)** Limit equilibrium analysis showing the global minimum slip surface and factor of safety for the simulated pre-collapse deposit with the effects of a water table 1 m above the contact between the pre-collapse deposit and the underlying bedrock.

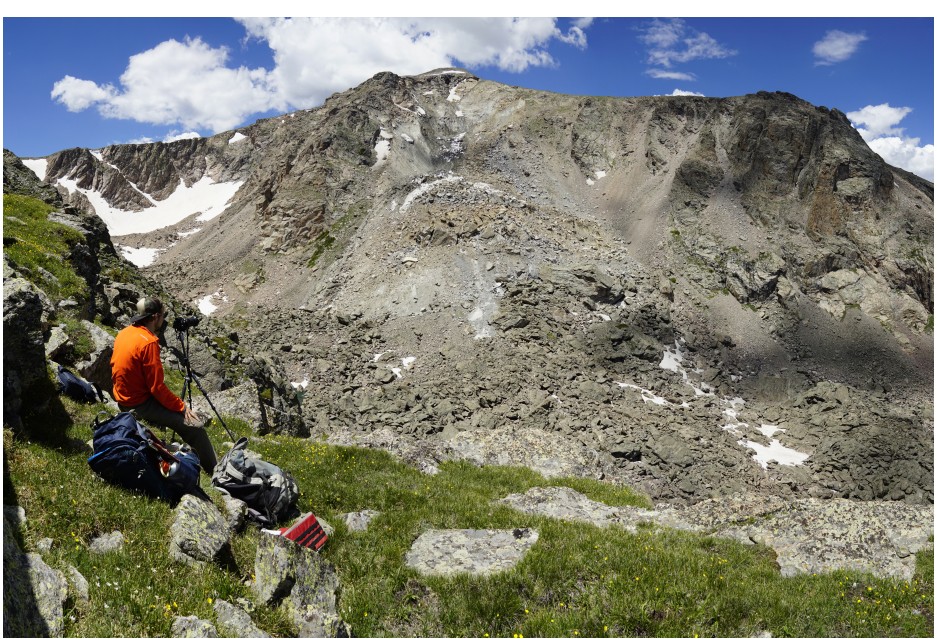

**Figure A7.** Photo of slide taken from south-southwest during SFM survey - one week after the collapse. Photo Credit: Benjamin Lehmann.

**Table A2.** Coordinates of the stations where the photographs were acquired.

| Station ID | x coord. | y coord. | z coord. (m a.s.l.) |
|:---:|:---:|:---:|:---:|
| 1 | 105°40.044′ | 40°17.8760′ | 3462 |
| 2 | 105°40.1335′ | 40°17.8441′ | 3471 |
| 3 | 105°40.2262′ | 40°17.8517′ | 3443 |
| 4 | 105°40.3847′ | 40°17.8460′ | 3447 |
| 5 | 105°40.4287′ | 40°17.8296′ | 3452 |
| 6 | 105°40.5145′ | 40°17.7625′ | 3547 |
| 7 | 105°40.5481′ | 40°17.7219′ | 3594 |
| 8 | 105°40.2899′ | 40°17.7264′ | 3576 |
| 9 | 105°40.1936′ | 40°17.7611′ | 3547 |

**Table A3.** Ground Control Point X, Y, Z, and Total errors in cm and error in SfM model px.

| Label | X error (cm) | Y error (cm) | Z error (cm) | Total (cm) | Image (pix) |
|-------|------------|------------|------------|----------|-----------|
| GCP1 | 0.387 | -12.626 | 17.858 | 21.874 | 10.849 (46) |
| GCP2 | -3.810 | 2.052 | -19.821 | 20.288 | 13.109 (49) |
| GCP3 | 7.790 | 8.976 | 9.717 | 15.352 | 4.105 (68) |
| GCP5 | -5.117 | -9.542 | -6.412 | 12.584 | 2.211 (91) |
| GCP4 | 0.686 | 11.160 | -1.124 | 11.237 | 3.680 (48) |
| Total | 4.517 | 9.589 | 13.028 | 16.795 | 7.296 |

*Author contributions.* Each author played a key role in this research project. MM guided the research from the start by bringing together the group and scheduling meetings and drafting this manuscript. He also contributed the climate analysis. BL and BC contributed the SfM modeling and change detection analysis - as well as the metrics of landslide mobility. GB and AH contributed the Insar and planet Image correlation analysis. BR contributed to the discussion editing of the final manuscript. LA contributed the slope stability modeling and aided MM in the climate analysis and permafrost modeling. AH contributed the Planet image correlation and aided the InSAR efforts. IO contributed the permafrost modeling. JM added the Google Earth based digital image correlation. All authors contributed to the editing of this manuscript.

*Competing interests.* The authors declare that they have no conflict of interest

*Disclaimer.*

*Acknowledgements.* The first author would like to thank Dr. Harrison Gray (USGS) for sending the initial video of this landslide posted to Twitter that spurred this research to action. Part of this research was carried out at the Jet Propulsion Laboratory, California Institute of Technology, under a contract with the National Aeronautics and Space Administration (80NM0018D0004). We would also like to thank the support of the French ANR-PIA funding program (ANR-18-MPGA-0006). We also thank the developers of the Digital Image Correlation code we implemented with Google Earth Imagery https://github.com/bickelmps/DIC_FFT_ETHZ.We also thank Tim Weinmann and Dr. Jill Baron for their time and fruitful conversations in the drafting of this manuscript. Thank you also to David O'Leary of the USGS Utah Water Science Center for his confidence and Dr. Katherine Dahm for your early support on this project.

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
