# Peer review of "Alpine hillslope failure in the western US: Insights from the Chaos Canyon landslide, Rocky Mountain National Park USA"

_EGUsphere, 2023_

## Referee Comment (RC2)

**Alpine hillslope failure in the western US: Insights from the Chaos Canyon landslide, Rocky Mountain National Park USA**

Matthew C. Morriss1, Benjamin Lehmann2,3, Benjamin Campforts2,4, George Brencher5, Brianna Rick6,7, Leif Anderson8, Alexander L. Handwerger9,10, Irina Overeem2, and Jeffrey Moore8

1Earth Sciences Department, University of Oregon Eugene, OR, USA 97403

2Institute of Arctic and Alpine Research, University of Colorado, Boulder, CO, USA

3Univ. Grenoble Alpes, Univ. Savoie Mont Blanc, CNRS, IRD, Univ. Gustave Eiffel, ISTERRE 38000 Grenoble, France

4Department of Earth Sciences, VU University Amsterdam, Amsterdam, 1081HV, the Netherlands

5Civil and Environmental Engineering, University of Washington, Seattle, WA, USA

6Department of Geosciences, Colorado State University, Fort Collins, CO 80523, USA

7Alaska Climate Adaptation Science Center, Fairbanks, AK, 99775, USA

8Department of Geology and Geophysics, Salt Lake City, UT, USA

9Jet Propulsion Laboratory, California Institute of Technology, Pasadena, 91109, USA

10Joint Institute for Regional Earth System Science and Engineering, University of California, Los Angeles, Los Angeles, CA 90095, USA

Correspondence: Matthew Morriss, PhD (matthew.c.morriss@gmail.com)

**Abstract.** The Chaos Canyon landslide, which collapsed on the afternoon of June 28th, 2022 in Rocky Mountain National Park presents an opportunity to evaluate instabilities within alpine regions faced with a warming and dynamic climate. Video documentation of the landslide was captured by several eyewitnesses and motivated a rapid field campaign. Initial estimates put the failure area at  $66,630 \text{ m}^2$ , with an average elevation of 3,555 m above sea level. We undertook an investigation of previous

- 5 movement of this landslide, measured the volume of material involved, evaluated the potential presence of interstitial ice/snow within the failed deposit, and examined potential climatological forcings at work in causing the collapse of the slope. Satellite radar and optical measurements were used to measure deformation of the landslide in the years leading up to collapse. From 2017 to 2019, the landslide moved  $\sim$ 5 m yr-1, accelerating to 17 m yr-1 in 2019. Movement took place through both internal deformation and basal sliding. Climate analysis reveals the collapse took place during peak snowmelt, and 2022 followed
- 10 10 years of higher than average positive degree day sums. We also made use of slope stability modeling to test what factors controlled the stability of the area. Models indicate even a small increase in the water table reduces the Factor of Safety to <1, leading to failure. Material volumes were estimated using Structure from Motion (SfM) models incorporating photographs from two field expeditions on July 8th, 2022 10 days after the slide. Detailed mapping and SfM models indicate ~ 1,258,000  $\pm$  150,000 m3 of material was deposited at the slide toe and ~1,340,000  $\pm$  133,000 m3 of material was evacuated from the
- 15 source area. Our holistic approach to the collapse of the Chaos Canyon landslide provided an opportunity to examine a landslide that may be representative of future dynamic alpine topography, wherein failures becomes more common in a warming climate.

**1 Introduction**

- The Chaos Canyon collapse took place during a sunny early-summer day at 3:31 PM local time on June 28th, 2022 in Rocky
  Mountain National Park (RMNP), Colorado, USA (Fig. 1). The event was first reported through social media from bystanders situated at Lake Haiyaha (Fig. 1) and directly beneath the slide. These social media posts resulted in a rapid response from Earth scientists who sought to investigate: 1) the causes of collapse; 2) the mechanisms triggering this collapse such as a landslide, the collapse of a rock glacier or the role of cryogenic processes in general, and 3) the amount of material mobilized by the collapse. More broadly, this investigation developed through a desire to understand whether this failure is part of the global
  trend toward cryosphere instability and the degradation of permafrost conditions driven by climate change (Geertsema et al., 2022; Patton et al., 2019). Mass movements in alpine regions have been documented with increasing frequency in the European Alps, Canada, and Alaska (Dai et al., 2020; Geertsema et al., 2022; Lacroix et al., 2022); however, few such observations have been made in the continental United States. RMNP was visited by ~4.4 million people in 2021, making it the 14th most visited National Park in the US (NPS, 2022). As the climate warms, developing awareness around future alpine hazards within RMNP
- 30 and across the broader western U.S. Cordillera is an important priority.

**1.1** Stability of alpine topography**

A warming climate has led to a warming of permafrost, glacier retreat, and growing instability in alpine regions across the globe (Patton et al., 2019; Shan et al., 2014). An increase in rock fall, landslides, and glacier instability has been well documented in the European Alps and the alpine regions of Canada and Alaska (Cossart et al., 2008; Deline et al., 2021; Kos et al., 2016;

- 35 Geertsema et al., 2022). Some of these events have received international news coverage (e.g. areas below the Planpincieux glacier in Italy on the flanks of Mont Blanc) and resulted in intermittent area closures and evacuations (Dematteis et al., 2021; Giordan et al., 2020). Other events have been more remote and garnered interest from the scientific community but have not had an impact on a broader population (e.g. Lipovsky et al., 2008). So far, few notable increases in instabilities in the alpine regions of the conterminous United States have been reported. It remains to be seen whether mountainous terrain in the conterminous
- 40 US will experience a similar increase in events like those seen in Alaska and the European Alps. The current geomorphological evolution of these alpine environments integrates both (1) the long-term response (i.e. on scales of 1-10 ka) to glacial retreat and glacial conditioning of the topography, and (2) the recent impact of a changing climate (i.e. on time scales of 10-100 years) via glacier retreat and permafrost degradation (Huggel et al., 2010; Stoffel and Huggel, 2012). Quantifying the dynamics of mountain landforms facing climate change is proving difficult and predictions of the geomorphological response such as
- 45 mountain slope instability to future climate scenarios remain limited because the processes involved operate on interdependent timescales and are therefore difficult to characterize. Herein, we seek to both develop the tools necessary to analyze those events and understand the stability of similar alpine slopes within a warming climate regime by characterising the relation between permafrost, topographic and climate forcings, and slope instabilities.

---

## Author Comment (AC1)

Response to Review 1

We appreciate the reviewers enthusiasm to see our work published in ESURF. Below we outline several responses to direct comments raised in their review. We were not provided line-by-line comments so we respond to these comments directly herein. Line-by-line comments will be provided for Reviewer #2 that asked for them.

—--

*"Although the authors mentioned that there are abnormal climatic factors before the final collapse of the landslide, there seems no quantified relation between climatic forcing and the landslide."*

We appreciate the reviewer's feedback, but we do not agree with the reviewer on this point. While in geomorphology we will always struggle to show a distinct difference between correlation and causation, our manuscript makes a strong argument for the impact of climate on this landslides' failure. Sixteen of the last 20 years had positive temperature anomalies as compared with the 30 year running average (Fig. 11). We were able to show from the Bear Lake Snotel data an increasingly warm average surface temperature through time. Moreover, our modeling efforts do indicate the potential presence of intermittent permafrost that will only continue to thaw with warming temperature. While this does not directly prove causation, we can only discuss how a warming climate leading to failure is one potential hypothesis that appears to have some evidence behind it.

—------

*I suggest the authors use models to simulate the process of the climatic factors (temperature induced permafrost thaw and precipitation) on the landslide.*
*There seems to be large gaps among techniques of deformation derivation, factor of safety modelling and SfM analysis. A better frame may be to use models to model landslide deformation processes with climatic inputs to analyze its mechanisms. Then use SfM to assess its consequences to erosion.*

Our approach was to be rooted in empirical observations. While we did take some time for modeling of permafrost and slope stability modeling, we opted to not pursue any in depth fine-scale modeling of the landslide and the factors affecting it. We prefer to have this paper be a mostly empirical paper and we may follow up on this effort by developing the model the reviewer suggests, but at this point, we do not think this makes sense given the data we have on hand. When we come to tackling a more direct model, we will incorporate the reviewers comments and the results of this study into such a model.

—--

*In addition, there seems to be little results derived from the InSAR.*

InSAR is only practicable for deformation rates of cm/yr, or more specifically if the phase change between images is greater than pi radians (~2.77 cm for our site) unwrapping will

fail (Itoh, 1982; Handwerger et al. 2015). Through our pixel tracking work, we were able to measure deformation of the Chaos Canyon landslide at the rate of meters/year. Such rapid rates of deformation lead to unwrapping errors in the InSAR results, making them highly suspect.  While we do discuss our attempts to use insar, we also provide information regarding the errors we found and why this technique will not be useable for further deformation analysis.

References:
Itoh, K. (1982). Analysis of the phase unwrapping algorithm. *Applied optics*, *21*(14), 2470-2470.

Handwerger, A. L., Roering, J. J., Schmidt, D. A., & Rempel, A. W. (2015). Kinematics of earthflows in the Northern California Coast Ranges using satellite interferometry. *Geomorphology*, *246*, 321-333.

---

## Author Comment (AC2)

**Summary**

In 'Alpine hillslope failure in the western US: Insights from the Chaos Canyon landslide, Rocky Mountain National Park USA', authors Matthew Morriss, Benjamin Lehmann, Benjamin Campforts, George Brencher, Brianna Rick, Leif Anderson, Alexander Handwerger, Irina Overeem, and Jeffrey Moore offer a comprehensive look at a large and rapid bedrock slope failure in the Rocky Mountains. The authors use a host of tools to better understand the morphology, kinematics, history, and triggering of the Chaos Canyon Landslide (CCLS), and argue that such events, which represent a potentially substantial and underappreciated hazard, may only become more common in the coming years. This manuscript offers a sort of blueprint for how to conduct reconnaissance studies of similar bedrock landslides occurring in alpine environments. However, despite all the methods thrown at this slide, the authors still struggled to outline why the CCLS failed catastrophically on June 28th, 2022. Some of this uncertainty lies in the inherent complexity of large, creeping bedrock landslides, where the morphology and stress fields change from season to season. Moreover, some of the methods used in this paper proved more useful than others. The authors present a fantastic structure-from-motion model and accompanying morphology assessment, including precise volume estimates. The optical image-correlation based displacement history offers tremendous insight to pre-catastrophic-failure landslide behavior, and the careful historical analysis of climate and snowmelt shed light on the possible triggering mechanisms, and on how anomalous spring, 2022 was compared to past years. However, the Insar analysis and slope stability model did not seem to add much to the overall understanding of the CCLS. It was unfortunate that the Insar analysis, which could have been quite useful, suffered from unwrapping issues. I go into more detail in the following 'main points' regarding the slope stability modeling, **but Slide2 is probably not the appropriate tool to study this type of slope failure, nor does the experimental setup of adding water to a slope with FS~1 broaden our understanding of the triggering mechanics**. Finally, as I expand on in the following comments, the manuscript would benefit from a more targeted and evidence-based discussion of how the CCLS fits into the broader realm of cryosphere-related geohazards. Despite these critiques, this manuscript offers an important look at a landslide which may have broad implications for future geohazards. I recommend publication in Earth Surface Dynamics after addressing the following main points and line-by-line edits in the attached commented pdf.

Sincerely,

Dr. Sean Lahusen

We would like to thank Dr. Lahusen for the time he took to review our manuscript. We also would like to thank the Editor for ensuring that we have a strong review process to help our paper be the best possible story. We have taken Dr Lahusen's suggestions into account and detail below our response to his broader comments. We also attach a longer PDF tracking changes, line by line, in response to Dr. Lahusen's specific feedback.

Best Regards,

Matthew Morriss & Ben Lehmann

**Reviewer main points**

1. **The manuscript would benefit from more specificity and cited evidence when describing the nature of the hazard that the CCLS represents, how it fits into the global and regional records of glacier and permafrost ice loss, and in what ways we can meaningfully extrapolate that hazard to the rest of the continental U.S.**

While I don't expect the authors to be able to definitively assign the cause of the CCLS, I struggled to come away from the manuscript with a better understanding of the most likely *primary* underlying cause – permafrost degradation, unusually high snowmelt rates, or more inherent changes to the landslide morphology and shear surface degradation? All the above*? Was anything exceptional about June 28th, 2022? We can assume that many other fluctuations in pore-water pressure in past years had initiated or accelerated temporary pulses of CCLS deformation, so we know the slope was not in a stable configuration to begin with.* And yet, these pulses of movement did not lead to catastrophic collapse in the past. A pertinent question seems to be: what was different on June 28th? If you could prove the hydrologic conditions were historically elevated compared to past summers, this could be a convincing argument. However, it seems like the snowmelt totals where actually on the lower end compared to the last 5 years (Figure 6B). Another hypothesis the authors discuss is that the shear zone was in fact not fully developed, but had been developing over many years, and finally reached a critically weak state where an unexceptional increase in pore-water pressure was now sufficient to trigger catastrophic collapse. This seems like a reasonable hypothesis. Similarly, perhaps the landslide, which was perched on a steep slope, simply crept itself into a new, inherently unstable geometry on the slope where catastrophic failure was possible. *Subtle geometric changes between the slide mass and the underlying slope can have a pronounced effect on landslide stability*.

We appreciate Dr. Lahusen's thoughts on the potential causes of the sudden collapse of the Chaos Canyon landslide. We've opted to modify our text in favor of Dr. Lahusen's comments: that potentially a decrease in permafrost sets the stage for an unexceptional increase in pore-water pressure combined with weakening of the shear plane and a continually less-stable landscape position for the slide lead to the June 28th failure. While we cannot discern exactly which of the several triggers may be most responsible for the failure, we have changed our text to represent these hypotheses.

If permafrost degradation is indeed the primary conditioning mechanism for this catastrophic collapse, then to argue that these types of events are likely to become more

common across the continental U.S. also requires some discussion of similar terrain in the continental U.S. How widespread is permafrost at sub-arctic latitudes? Obu et al. (2019) discusses this some, while also citing the earliest study of permafrost in the Rockies that I could find (Ives and Fahey, 1971):

*'All permafrost zones occur in the Rocky Mountains of the USA with sporadic permafrost in Colorado at elevations above 3200 m asl and discontinuous permafrost above 3500 m asl, results which agree with observations by Ives and Fahey (1971). Isolated permafrost patches are modelled in the highest peaks of the Sierra Nevada in the southwestern USA, which corresponds well with the presence of active rock glaciers in the area (Liu et al., 2013)'*

Thank you for this comment. We've added these reference and a specific reference to the elevations where permafrost may be found in Colorado and the SW US.

A major point of this paper is that events like the CCLS are only going to become more common. You mention a transformational period in alpine landscapes over the last few decades and cite 'changes to the ice glaciers of the world' – but what changes are we talking about, specifically - and what rates? This message comes off as overly-vague. Replacing some of this vague language with targeted evidence would be much more compelling. The global cryosphere has undergone enormous changes in the last 115kya, including numerous glacial advances and retreats in the last 20kya. Globally, most glaciers have been receding since the end of the LIA in the middle 19th century. Undoubtedly, some of the presently-observed glacial retreat rates, permafrost degradation, and increases in alpine rock avalanche activity are due to anthropogenic climate change over the last century. However, if you want to argue the last few decades are transformational, I would reference some of the following work: *Marzeion et al. (2014) show that only 25% of global glacial mass loss from 1851-2010 was due to anthropogenic causes – but this percentage nearly tripled to 69% for the period of 1991-2010. Hugonnet et al. (2021) show a dramatic acceleration in global glacial ice mass loss and thinning from 2000-2019*. Furthermore, Christian et al. (2018) suggest that much of the effect of the last few decades of warming has already been baked into mountain glaciers in the form of 'committed retreat', which will result in continued dramatic glacial terminus retreat, even in the absence of additional warming. *Cite more specific evidence and rates of this transformational period of change over the last few decade, because the idea of a transformational period strongly suggests we have left a long period of glacial and general cryosphere stability*. If so, when was this period of stability – the end of the LGM? Following retreat from the Younger Dryas? The LIA (where many glaciers in saw substantial advances and subsequent retreats)? Another way I would think about this from a geomorphological perspective is: when was the last time a landscape experienced ice-free conditions?

We thank the reviewer for this comment. We have specifically added text and appropriate references to the introduction and discussion to reflect this comment. Citing the increased rate of glacial retreat brought to our attention by the reviewer and going further, looking into alpine

rockfall rate inventories (Ravanel and Deline 2011; Deline et al. 2021). These references build a narrative of accelerating destabilization in alpine regions, with many of the changes driven by warming of the past few decades yet to be seen (e.g.. Christian et al., 2018). There are comparably fewer papers on the rate of change within the Coterminous United States, which is aa key argument of our paper, so many of the additional references and examples are from the Alps. We also added a caveat that little is known about the alpine landscape's stability in the RMNP region beyond the fact that there was a small Little Ice Age glacial advance (Benson et al., 2007).

Benson, L., Madole, R., Kubik, P., and McDonald, R.: Surface-exposure ages of Front Range moraines that may have formed during the Younger Dryas, 8.2calka, and Little Ice Age events, Quaternary Science Reviews, 26, 1638–1649, https://doi.org/10.1016/j.quascirev.2007.02.015, 2007.

Deline, P., Gruber, S., Amann, F., Bodin, X., Delaloye, R., Failletaz, J., Fischer, L., Geertsema, M., Giardino, M., Hasler, A., Kirkbride, M., Krautblatter, M., Magnin, F., McColl, S., Ravanel, L., Schoeneich, P., and Weber, S.: Ice loss from glaciers and permafrost and related slope instability in high-mountain regions, in: Snow and Ice-Related Hazards, Risks, and Disasters, Elsevier, 501–540, https://doi.org/10.1016/B978-0-12-817129-5.00015-9, 2021.

Ravanel, L. and Deline, P.: Climate influence on rockfalls in high-Alpine steep rockwalls: The north side of the Aiguilles de Chamonix (Mont Blanc massif) since the end of the 'Little Ice Age,' The Holocene, 21, 357–365, https://doi.org/10.1177/0959683610374887, 2011.

*Finally, perhaps it's not appropriate to lump all cryosphere processes together – mountain glaciers like those that carved chaos canyon are inherently dynamic systems. Permafrost – maybe less so.* How does this dynamic glacial history compare with the record of permafrost-underlain landscapes over the same timescale? Where are the changes in the last few decades most pronounced (especially in the lower 48, as this seems to be a main point of the paper)?

Thank you for this comment. We have been challenged to find equivalent studies of permafrost instability in the Lower-48 that we can point to showing rapid changes within the past several decades. Again, with so many publications coming from the Alps, our permafrost modeling, and our climate analysis show such changes are underway or perhaps will soon be underway due to the "baked in" behavior described previously. We have added one reference, O'Connor and Costa (1993), which is one of the few published works we could locate describing the hazards posed by alpine regions from a warming climate with mention of the lower-48. The dearth of papers on such changes in permafrost in the lower-48 and their associated hazards is compounded by the fact that the lower latitude and lower elevation regions that have sporadic or discontinuous permafrost will see the most rapid changes (Patton et al., 2019; Slater and Lawrence, 2013). But perhaps the lack of such literature is not a sign that the phenomena does not exist. We continue to frame our discussion and introduction that there is evidence in the Alps, Alaska, and the Canadian Rockies of increasing instability, but it remains to be seen if the lower-48 will see large scale and rapid changes. The evidence we present appears to indicate that

the Chaos Canyon landslide is an example of this type of instability, perhaps one of the earlier landslides recorded in the lower-48 that appears linked to climate change.

Patton, A. I., Rathbun, S. L., and Capps, D. M.: Landslide response to climate change in permafrost regions, Geomorphology, 340, 116–128, https://doi.org/10.1016/j.geomorph.2019.04.029, 2019.

O'Connor, J. E. and Costa, J. E.: Geologic and hydrologic hazards in glacierized basins in North America resulting from 19th and 20th century global warming, Nat Hazards, 8, 121–140, https://doi.org/10.1007/BF00605437, 1993.

Slater, A. G. and Lawrence, D. M.: Diagnosing Present and Future Permafrost from Climate Models, Journal of Climate, 26, 5608–5623, https://doi.org/10.1175/JCLI-D-12-00341.1, 2013.

1. **Mohr-Coloumb limit equilibrium slope stability models like Slide2 seem insufficient to model a creeping, accelerating rock mass**

Landslides like the CCLS, which have been creeping intermittently for years with total displacement >10 meters, probably have pronounced strain-rate dependent behavior. Ideally, a slope stability model could account for (or at least approximate) this strain rate dependency, in order to capture the interaction between some of the following landslide characteristics:

1. Rate-hardening or rate-weakening frictional properties of the shear zone material
2. Material dilatancy vs contraction of the shear zone material
3. Evolution of drained vs. undrained conditions along the shear zone
4. Changing landslide geometry (>10 meters of displacement since the landslide initiation)
5. External factors influencing pore-water pressure, like snowmelt

Of all these important factors, Slide2, which is typically used to model the failure of intact rock slopes, can only model the last. Of course, no model can totally capture the intricacies of a natural slope, and even if it could, ascertaining these physical properties for the CCLS may prove impossible given the lack of instrumentation prior to failure. Still, in my opinion, *Slide2 is too much of an oversimplification for this very complex slope failure, and is not the appropriate tool for modeling a creeping, rate-dependent bedrock landslide - especially if interstitial ice was involved*. GLE models like Slide2 typically predict a binary outcome: total stability (no slope deformation whatsoever) or catastrophic failure. This CCLS does not exemplify this use case - in all likelihood the CCLS was creeping right before it collapsed. I can see how the authors tried to replicate this condition in Slide2 by

using a realistic, empirically-derived geometry and tuning the model parameters such that the FS was just barely greater than 1.0, representing a slope on the verge of collapse.

However, a slope with a FS~1 in Slide2 does not represent an intermittently creeping landslide – it represents an intact bedrock slope that has experienced no displacement. *Most problematically, the addition of pore water pressure (in the form of any amount of additional water perched above the slip surface) will invariably push a modeled slope with a calculated FS of ~1.0 into a state of instability*. I don't understand what value such an analysis adds to our understanding of the CCLS. If the authors could somehow prove the slide was *particularly* sensitive to small perturbations in pore-water pressure, this may be a more compelling analysis. For instance, you could add the same amount of water to the slip surface as the modeled snowpack melt volume (although this would bring up the question of why previous years with higher snow melt rates did not trigger the catastrophic collapse). To be clear, after reading your manuscript I am convinced that elevated pore-water pressure, likely from snowmelt, played an important role in triggering the June 28[th] collapse, but Slide2 is an inappropriate tool to study this type of landslide and does not reinforce an otherwise compelling assertion.

We would like to thank the reviewer for his detailed response and suggested revisions to this section of our text. The reviewer is correct in laying out several landslide characteristics there were not tested nor evaluated in our manuscript; however, this landslide provided very little opportunity to construct the models that the reviewer describes. Our manuscript is built on simple empirical observations and models that incorporate those observations. We have very little data about the pre-failure landslide. Moreover, to build the model that is recommended, evaluating rate-hardening/weakening and evaluating dilatancy/contraction would require more physical parameters than we currently have available about the state of the landslide pre-collapse, or as the reviewer put it so well: "ascertaining these physical properties for the CCLS may prove impossible given the lack of instrumentation prior to failure."

Our approach with modeling is to explore the potentials of the landslide system, as opposed to trying to build a complex representation of the system. That is not to say either approach is more correct, but for the information we have readily available (i.e. the slide geometry in 2017 and its post-collapse geometry) a simpler model is an easier to understand entry point for the discussion of the collapse of the Chaos Canyon landslide. We would welcome and potentially collaborate on future modeling efforts to achieve the more holistic representation of the system described by the reviewer. This would likely be a research project and manuscript on its own right. However, an investigator seeking to answer these questions with a more complex model will continue to be confounded by the lack of available physical parameters pre-collapse; perhaps there are other landslides that are more data rich and provide greater opportunity to learn about the complex systems at work.

Including the Slide2 analysis provides an opportunity to highlight the sensitivity of this system to increases in pore fluid pressures an important discussion point for our manuscript. We prefer to keep this analysis in the manuscript as it is. To add more details would require more physical parameters we do not currently have access to nor ability to estimate in a robust manner. To do so would detract from the goal of keeping our methods rooted closely to our empirical observations.

1. **The paper suggests tackling climate-driven alpine slope stability hazards on a national scale, but offers no new related data or analyses outside of the Chaos Canyon Landslide**

I think the scope of this paper in its current form is totally sufficient and represents a fantastic addition to our understanding of an important landslide - potentially a harbinger of future, similar events. Unfortunately, the authors also suggest they intend to examine similar processes at the scale of the continental U.S. The author's stated goals for this study are to: 1.'Develop the tools necessary to analyze those events' and 2. 'Understand the stability of similar alpine slopes within a warming climate regime by characterizing the relation between permafrost, topographic and climate forcings, and slope instabilities.' (lines 46-48). The manuscript title is also suggestive of this broader national-level analysis. While it is totally reasonable and very intriguing to speculate about extrapolating the implications of this single landslide to a national scale, this study does not offer any new analyses at that scale. Original data and analyses in this paper are confined to the CCLS. So, *while such speculation is an important element of the paper's discussion that should remain, it does seem like the paper promises but then does not deliver on new analyses or data pertinent to the broader hazard. I suggest subtle rewording throughout the manuscript to reflect this*.

We appreciate the reviewers comment on this point and have adjusted both our title and the introduction of our manuscript to match this feedback. While the discussion remains broad, the introduction now focuses more on the Chaos Canyon Landslide as a case study from which we can test techniques and learn about alpine landform stability, with the hope that these suite of tools, ideas, and models can be used elsewhere. It is our hope that these comments are in lines with the reviewer's comments and make the story we are trying to tell overall more consistent.

**References Cited**

Christian, J. E., Koutnik, M., & Roe, G. (2018). Committed retreat: controls on glacier disequilibrium in a warming climate. Journal of Glaciology, 64(246), 675-688.

Obu, J., Westermann, S., Bartsch, A., Berdnikov, N., Christiansen, H. H., Dashtseren, A., ... & Zou, D. (2019). Northern Hemisphere permafrost map based on TTOP modelling for 2000–2016 at 1 km2 scale. Earth-Science Reviews, 193, 299-316.

Hugonnet, R., McNabb, R., Berthier, E., Menounos, B., Nuth, C., Girod, L., ... & Kääb, A. (2021). Accelerated global glacier mass loss in the early twenty-first century. *Nature*, *592*(7856), 726-731.

Zemp, M., Huss, M., Thibert, E., Eckert, N., McNabb, R., Huber, J., Barandun, M., Machguth, H., Nussbaumer, S.U., Gärtner-Roer, I. and Thomson, L., 2019. Global glacier mass changes and their contributions to sea-level rise from 1961 to 2016. *Nature*, *568*(7752), pp.382-386.

Bolch, T., Kulkarni, A., Kääb, A., Huggel, C., Paul, F., Cogley, J.G., Frey, H., Kargel, J.S., Fujita, K., Scheel, M. and Bajracharya, S., 2012. The state and fate of Himalayan glaciers. *Science*, *336*(6079), pp.310-314.

Marzeion, B., Cogley, J.G., Richter, K. and Parkes, D., 2014. Attribution of global glacier mass loss to anthropogenic and natural causes. *Science*, *345*(6199), pp.919-921.

Line by Line Response to Reviewer 2

Line 23: These questions should be consistent with those outlined in section 1.3

*We have changed the listened investigation items to match section 1.3*

Line 28: I would add a transition sentence here, something along the lines of: 'Despite the lack of observed climate-driven landslides in the continental U.S., these slope failures pose a potentially high risk to the ever-increasing number of people who spend time in alpine environments.'

*We have added a transition sentence, per the reviewer's recommendations*

*Line 35:* What does 'glacier instability' mean? Serac collapse? Ice avalanches from hanging glaciers/icefalls? Full translation sliding failures driven by basal pore-water increases like those documented by Kaab et al. (2018)? It could also be interpreted to mean glacier down-wasting and terminus retreat (i.e. the position and mass of glacial ice is unstable over time). So, I'd clarify. If this term is well-known in the community, disregard this comment.

*We have further defined 'glacier instability' and actually remove that phrase from the manuscript. Instead specifying that we mean glacier retreat or changes in glacier mass globally.*

Line 41: I'm a bit confused by this timescale. Glacial conditioning of alpine landscapes in the cont. US has been occurring for far more than 10,000 years - cyclically throughout much of the Pleistocene, correct? If this is in reference to the amount of time a landscape has experienced 'ice-free' conditions since glacial retreat from the last maximum, wouldn't it be more like 15-20 ka for much of the PNW?

*The manuscript now specifies that we are talking about landscape evolution combining long timescale (10s to 100s of ka) in addition to recent climate change. This should stratify the reviewers comment.*

*Line 46:* what events? Assuming alpine bedrock landslides but it's not stated

*We specify now in the text we mean high elevation, mid-latitude landslide here.*

*Line 45:* Very nice figure

*Thank you!*

*Line 45:* I would state that the imagery predates the catastrophic failure of the CC LS

*We have changed the caption to say so*

*Line 50:* A general comment: throughout the manuscript, I would find a way to be more clear about differentiating the June 28th rapid failure vs. the existing landslide that had been deforming for years.

*Yes, we recognize that this is a bit confusing. We've done our best to tighten up the language and make it more clear the difference between the pre-existing landslide landform and the rapid failure on June 28th.*

*Line 54:* What deposit? This is the first we have heard of this. Assuming not the landslide deposit...? Could be as easy as changing 'The deposit' to "A deposit'?

*Fixed*

*Line 127:* This term sounds self-explanatory, but a brief description would still be useful.

*We've added one!*

*Line 210:* as someone not familiar with data processing for InSAR, I would appreciate a citation referencing this issue

*References added!*

*Line 232:* I almost wonder if it's worth moving this figure entirely to the supplemental materials (or at least panels B-D). It's unfortunate that the InSAR did not provide more useful insight, but it doesn't seem worth having a figure in the main text that essentially shows the technique wasn't very useful in this particular application.

*We've kept this InSAR figure within the main text but modified it to be a 2 row, 2 column figure for efficiency of space.*

*Line 237:* Add scale to panel c. Also, just a suggestion, not totally necessary, but I would change the tick mark division values to be less random, i.e. 1600,3200,4800 instead of 1638,3276,4914. Feel free to ignore if this is difficult to do.

*This figure was created using pixels as the coordinate system, so we aren't totally able to change the frame coordinates easily, but we did add a scale to the first panel and a reference to the pixel scaling in the caption..*

*Line 257:* True, but compared to the past 5 years, it was average or below-average. Figure 6b indicates that 2018,2020, and 2021 had reached greater snowmelt by June 28th than 2022. This seems to stand at odds with any argument that June 28th, 2022 was exceptional

*We've changed the text to indicate that yes 2022 was an unexceptional year. Hopefully this is more clear to the reader now.*

*Line 276:* should be degree symbol?

*Fixed*

*Line 278:* Cool model and very clear, readable figure

*Thanks!*

*Line 282:* should this be meters?

*Fixed*

*Line 290:* I'm not convinced this analysis does prove that. It simply proves that Slide2D works as advertised - if you reduce the effective normal stress by ANY amount in a modeled slope, of course the FS will be reduced by some amount. If this exercise is done for a slope with a modeled FS barely above 1.0, it will be reduced to below 1.0. This analysis unfortunately does not inform the relative sensitivity of the slope to pore-water pressure perturbations. See an extended discussion in the 'Main Reviewer Points' section of my review.

*We have responded to this comment in the Main reviewer points attached above.*

*293:* cite

*Fixed!*

*299:* I think this is an important point that deserves some discussion. Why do the authors think no similar events have occurred in the lower 48? Is the CCLS exceptional or have we only recently surpassed some threshold related to warming of alpine environments? Or some other reason?

*We have expanded on this comment from the reviewer and added another section to the discussion postulating on why there are fewer to no observed or discussed high elevation landslides in the lower 48 related to climate change at this point.*

*Line 313:* Glad to see a discussion of this phenomenon here. The same observed, accelerating landslide behavior could also be caused by shear zone development, regardless of whether the material itself is rate-weakening or rate-hardening on an particle scale.

*Thanks! We appreciate the positive feedback..*

*Line 326:* would this technically make it a rock glacier?

*Potentially… Although the deposit lacks many of the typical morphological characteristics common to a rock glacier. We discussed this at some length as a group and opted to call the deposit/landform a landslide rather than a rock glacier.*

*Line 336:* I'm a little confused here. The active layer was only modeled to be ~2 meters deep in a total thickness of 15 m of permafrost. Having trouble reconciling how this relatively shallow active layer influenced flowpaths to such depths as the slip surface.

*We have tightened our language a bit. The general idea is that if there's a consistent permafrost horizon across the landslide then snowmelt will not be able to penetrate to the failure surface; however, if that permafrost becomes discontinuous it may be possible for new flowpaths to open and melt can now penetrate the landform to the failure surface. There is also the potential for fissures to open due to the sliding of the landslide.*

*Line 360:* It seems like a major point you're trying to make in this paper is that events like the CCLS are only going to become more common. But, this message comes off as overly-vague. See reviewer main points for more details on this.

*We have addressed this review in the main comments section of the reviewer comments higher up.*

*Line 373:* Given the troubles with the InSAR analysis, any advice for future studies?
*We recommend that insar be used in concert with pixel tracking and tried to make that clear in our language.*

*Line 375:* See main reviewer points on this - but I'd like to know how common such locations are in the lower 48

*We have responded to this comment in the main reviewer comments*

*Line 387:* might note that a similar failure with higher ice content may travel further - representing an even greater risk to park visitors

*We thank the reviewer for this insight and modified our text to incorporate it into the conclusions.*

*Line 388: What years?*

*Fixed!*

---

## Author Comment (AC3)

We would first like to thank Dr. Tye for the time put forth in his careful review of our manuscript. As a third reviewer, we found some of his comments built on those from our first and second review but also caught important areas of improvement that speak to Dr. Tye's expertise. Below, we outline our direct responses to each of these comments and our improvements and changes. The final manuscript will be submitted separately to the journal editor for final editorial review.

The sans serif font in blue are Dr. Tye's comments The Serif font in black are our responses.

The manuscript by Matthew Morriss et al. provides an interesting case study of the Chaos Canyon Landslide (CCL) in Rocky Mountain National Park, including the character of the slope failure and its drivers. The manuscript is valuable for its application of a range of observational and modeling techniques to understand the event. The application of a wide range of techniques helps the authors to develop a comprehensive picture of this bedrock landslide, including the evolution of pre-failure creeping, and the volume of the slide mass. The authors explore the connection between the CCL and climate change, a topic of scientific and land-management interest. The authors generally do not overstate the significance of their results, which can only be speculatively connected to a climatic driver, but the manuscript would benefit from more critically assessing the potential mechanisms by which warming could have caused the CCL event. In addition to this point, I have several questions about the methods used and how the results are integrated, although I find no problems that jeopardize the validity of the authors' conclusions. In general, the manuscript presents a valuable case study and methodology that I believe will be of interest to the community and is appropriate for publication in ESurf after revision.

The weakest point of the manuscript is the connection between climate change and a CCL of the trigger. The occurrence of the CCL near the hottest time of the year is compelling in suggesting (speculatively) that temperature played a role in triggering the event. What is less clear is why the CCL occurred in the year that it did, which prevents clear establishment of a climate change-related mechanism. There is not a straightforward relationship between the behavior CCL and annual temperature, as 2022 was not an exceptionally warm year compared to the previous ~10 years, and the data do not resolve when pre-failure creep began. The authors explore both a reduction in interstitial ice and a meltwater-induced rise in groundwater as possible mechanisms for triggering the CCL. The hypothesis of significant interstitial ice reduction seems inconsistent with the authors' permafrost models, which indicate a maximum melting depth of <2 m in Summer 2022, an order of magnitude less than the depth of erosion (and thus minimum slip plane depth) indicated by the SfM analysis. Thus, the vast majority of the slide mass would still have been subject to freezing temperatures immediately before failure. The meltwater hypothesis is shown to be feasible through a simple factor of safety analysis, although the data and analyses presented don't establish how the magnitude and rate of meltwater produced in the area are likely to have fluctuated over the years, preventing assessment of any temporal trends that might explain the timing of CCL failure. It would be useful to see curves of annual precipitation and/or modeled meltwater production over time, if possible. Of course, pre-failure deformation of the CCL mass may have contributed to the timing of failure more than the specific conditions of 2022, but it is difficult to attribute this pre-failure activity to climatic forcing without better constraints on when it began. I don't see this issue as a fatal flaw for the manuscript, and the authors generally do a good job of

stating that a climate forcing mechanism for the CCL is speculative. However, given the interest in this topic, I think the manuscript would be enhanced (and made more impactful) by a more in-depth discussion of potential climatic forcing mechanisms.

I have outlined some additional significant but less important points that would benefit from clarification below, in no particular order, with line edits following. In addition to these points, the manuscript would benefit from a close rereading to identify typographic errors, ensure that figures are consistent with their captions, revisit the order of figure calls, and ensure that all figure panels are referenced in the text. The authors should also consider making the field photos, analyzed satellite imagery, and SfM model available for the sake of reproducibility.

We would like to thank Dr. Tye for his thoughtful review of our manuscript. Many of his discussion points touch on similar issues that were raised by our second reviewer. I have outlined more detail responses below. Where necessary, I will refer to our responses to our second reviewer and how we modified our manuscript appropriately.

*The weakest point of the manuscript is the connection between climate change and a CCL of the trigger. The occurrence of the CCL near the hottest time of the year is compelling in suggesting (speculatively) that temperature played a role in triggering the event. What is less clear is why the CCL occurred in the year that it did, which prevents clear establishment of a climate change-related mechanism. There is not a straightforward relationship between the behavior CCL and annual temperature, as 2022 was not an exceptionally warm year compared to the previous ~10 years, and the data do not resolve when pre-failure creep began.*

This comment is very similar to the feedback received from Reviewer No. 2. 2022 was an unremarkable year for snowmelt as we showed in figure 6B. In some ways, it's remarkable that the collapse *did* occur in that year as opposed to past years. In the newly revised text, we now discuss the potential for multiple confounding factors that may have built through time: 1) decrease in permafrost that allows for more open pore spaces for water to infiltrate into the landslide; 2) weakening of the shear plane beneath the landslide, and 3) a continually less stable landscape position as the landslide deforms and translates into a steeper and steeper position above Chaos Canyon. These three factors allowed for an unremarkable amount of snowmelt in 2022 to raise the pore-fluid pressure to a point that helped push the landslide over into catastrophic failure. We cannot ascertain when the creeping movement of the landslide initiated, so we're unable to further comment on the relationship between on the longer lived motion of the landslide and climate change; however, we do draw upon other literature examples of mass wasting and connections with climate warming to posit that warming likely plays a role in the initiation and continued movement of this landslide. It is this movement that sets the stage for its ultimate collapse in 2022.

*The hypothesis of significant interstitial ice reduction seems inconsistent with the authors' permafrost models, which indicate a maximum melting depth of <2 m in Summer 2022, an order of magnitude less than the depth of erosion (and thus minimum slip plane depth) indicated by the SfM analysis.* Thus, the vast majority of the slide mass would still have been subject to freezing temperatures immediately before failure.

We appreciate the reviewers comment here regarding the presence of permafrost. It appears to have been unclear in our manuscript and we've now changed some of our text to specifically mention that this modeling does not necessarily mean there is permafrost across the entire landslide, but that there is the potential for intermittent permafrost. Factors such as snow accumulation on the upper portions of the landslide that would make permafrost formation more difficult to form and persist were not included due to lack of reliable data. This model served to indicate that there is the potential for internal and interstitial ice in the landslide deposit. We also showed an increasing temperature anomaly at the landslide site through time, which would lead to an increased instability of any potential permafrost that may be there. The ~1 m of depth for the thaw front on the date of collapse, provides support for the idea that by June 28th, melt water would have been able to penetrate the slide deposit and makes it way through the slide material by flowpaths no longer occupied by interstitial ice. Melt water would also introduce another pathway for melt of interstitial ice by more efficiently conducting warmer water into the slide (e.g. Vedie et al., 2011).

Additionally, geotechnical testing of soils that experience freeze thaw also indicate that even compacted soils that experience regular freeze thaw have a higher hydrologic permeability (Kim and Daniel, 1992; Qi et al., 2006; Vedie et al., 2011). So even as permafrost in places may serve as an obstacle to infiltration, the active freeze thaw processes provide more pore spaces in the shallow surface for snowmelt to infiltrate and travel along the landslide potentially following flow paths deeper into the slide.

In response to this review and the comments from Reviewer No. 2, we have modified the text of our manuscript.

References cited:

Kim, W.-H. and Daniel, D. E.: Effects of Freezing on Hydraulic Conductivity of Compacted Clay, Journal of Geotechnical Engineering, 118, 1083–1097, https://doi.org/10.1061/(ASCE)0733-9410(1992)118:7(1083), 1992.

Qi, J., Vermeer, P. A., and Cheng, G.: A review of the influence of freeze-thaw cycles on soil geotechnical properties: Freeze-thaw and Soil Properties, Permafrost Periglac. Process., 17, 245–252, https://doi.org/10.1002/ppp.559, 2006.

Vedie, E., Lagarde, J.-L., and Font, M.: Physical modelling of rainfall- and snowmelt-induced erosion of stony slope underlain by permafrost, Earth Surf. Process. Landforms, 36, 395–407, https://doi.org/10.1002/esp.2054, 2011.

The meltwater hypothesis is shown to be feasible through a simple factor of safety analysis, although the data and analyses presented don't establish how the magnitude and rate of meltwater produced in the area are likely to have fluctuated over the years, preventing assessment of any temporal trends that might explain the timing of CCL failure.  It would be useful to see curves of annual precipitation and/or modeled meltwater production over time, if possible

We appreciate this comment and believe the text may already address it. The cumulative snowmelt curves are shown in Figure 6 Panel B going back nearly 30 years. I highlighted the individual last 5 years (including 2022). As was discussed in the response to Reviewer 2, 2022 was not an exceptional year from a melt volume perspective. It seems more likely that the previous years of movement aided in the development of a more distinct failure surface along with a weaker landscape position for the overall landslide deposit. Given the comment from the reviewer, Figure 6B appears sufficient. Moreover, the cumulative melt was calculated using the PDDS factor with a linear scaling, so the further temperature analysis displayed in Figure 11C is indicative of the temperature anomaly in preceding years.

Of course, pre-failure deformation of the CCL mass may have contributed to the timing of failure more than the specific conditions of 2022, but it is difficult to attribute this pre-failure activity to climatic forcing without better constraints on when it began. I don't see this issue as a fatal flaw for the manuscript, and the authors generally do a good job of stating that a climate forcing mechanism for the CCL is speculative. However, given the interest in this topic, I think the manuscript would be enhanced (and made more impactful) by a more in-depth discussion of potential climatic forcing mechanisms.

Given this comment from both Reviewer 2 and 3, we have bolstered our introduction and discussion sections with more text, and references, that catalogue the potential climatic mechanisms potentially at play in this landslide. We fully acknowledge, as the reviewer points out, that we cannot know when the failure began, which limits our insights into the slide at this time.

**Other points**

1. Composition of the CCL mass. The text is somewhat ambiguous as to whether the pre-failure mass was bedrock or regolith. Section 1.2 states, "The slide occurred along the contact between the Middle Proterozoic Silver Plume Granite and the early Proterozoic biotite schists." Does this mean that the slip plane is inferred to be the contact between these two units, or only that this is the geographic location where the slide occurred? The foliations mapped on Fig. A1 have dips similar to or less than the stated pre-failure surface slope of 40 degrees, consistent with CCL slip along a pre-existing foliation plane. Whether the pre-failure CCL material was bedrock or unconsolidated sediment would have implications for the failure mechanism—interstitial ice is probably less significant in igneous & metamorphic bedrock than sediment and foliation planes in bedrock might provide conduits for meltwater transport, so it is worth being more explicit about the composition of the slide mass. If the bedrock is important, consider adding the geology to Figure 2.

Thank you for this comment. It's possible that our description of the slide in the Introduction was not thorough enough or clear enough. We have revised the introductory text to make it clear that the landslide deposit which collapsed on June 28th, 2022 was a diamicton, or a deposit of poorly sorted material ranging in size from fine sediment too coarse boulders. Additionally, we have added The geology figure as Figure 2, highlighting the potential importance of the foliation. The text now reflects the potential contribution from a dipping foliation to accentuating the failure surface.

2. Intercomparability of the image correlation results. The image correlation techniques have an important role in the study, producing the only results that establish pre-failure movement of the slide material. Because of this, it would be valuable to see the image correlation results presented more systematically, including having similar figures for the different approaches taken with the Google Earth and PlanetScope imagery, such that the reader could evaluate the consistency of the results from the two imagery sources and distinct methods.

We appreciate the reviewers thoughts here; however, it's important to consider that these methods are collected from different satellites and from different points in time. We chose to keep this discussion of these two methods separate to reflect these differences and not confound the two separate methods for both collecting the data but also measuring deformation. We used the two methods in conjunction to best utilize their individual strengths. The Google Earth approach had very limited imagery options (just one pair of images to use) however the spatial resolution turned out very well for that image pair. The Planet imagery had many more options available meaning we could do temporal tracking, but the spatial resolution in each case was generally poorer than the Google Earth comparison. So each method gives something unique and complimented results from the other. We have added a figure that shows a direct comparison between the two methods to our supplement - per the reviewers recommendation.

3. Structured residuals in the SfM model. The difference model between the post-slide SfM model and the pre-slide topography (Figure 8) shows coherent differences outside the slide area. Areas downslope from the slide have negative differences and areas higher on the slope outside the slide have positive differences. I wonder if this reflects distortion in the SfM model, problems with registration to the DEM, or something else. This should be addressed in the text, along with any implications for the eroded and deposited volume estimates

The variations outside of the slide area are within the range of -5 to -15 meters downstream and +5 to +15 meters upstream (represented by light red and light blue colors in Figure 8). Your concern is valid as we've calculated an uncertainty of approximately 2.39 meters for this difference in the reference area (shown as a dashed polygon in Figure 8). We have three points of response to your comment:

1) The areas mentioned are distant from the mass movement, and the approach we used to center the terrestrial photogrammetry analysis might have introduced distortions away from the center of the landslide. This argument holds more weight when considering uncertainties upstream. Notably, the presence of a -5 to 5 meter margin around the landform provides reassuring evidence for higher accuracy within the landform itself.

2) The uncertainties arising from the terrestrial photogrammetry process result in an underestimation of volume differences, which has a somewhat conservative impact on our volume estimates.

3) The uncertainties could stem from the lack of Ground Control Points (GCPs) in the lower part of the area, downstream from the landform. a suitable GCP in this region couldn't be identified due to: (i) its proximity to the landslide, which lowered our confidence in selecting reference points in this area that were not affected by the movement, and (ii) its closeness to the scanning area, which led to occlusion of some potential bedrock surfaces by foreground relief.

**Line edits**

54-55 – reformat citation

I believe this is correctly formatted for the journal. I will make sure to check with the copy editors on this.

73 – redefine 'SfM' as this is its first use in the body of the paper

Fixed

125 – how was the environmental lapse rate calculated?

We have provided more details.

141 – how were the 305 photos collected from the 9 photo points (e.g., mosaic from each photo point location)?

These photos were individual frames collected with different shutter closures of the camera in use. There were a variety of number of photos taken from each site. They were then, as we describe in the text mosaiced using AgiSoft Metashape.

238-239 – rephrase to communicate greater confidence/reproducibility, e.g., "independent measurements of displacement of large boulders identified visually in the images are consistent with displacement magnitude inferred from image analysis"

Fixed to be consistent with the reviewers comment.

249 – change "1/velocity" to "inverse-velocity" or similar

Done

253 – rephrase "out of the ordinary" to "atypical" or similar

Done

263 – reintroduce what is being shown in this difference map and how it was obtained using a new topic sentence

Done

277-278 – revisit for syntax, degree symbol

Done

293 – add citations

Done

315 – I suggest eliminating the first clause as it is very different from where the paragraph is going

Done

346 – it appears that Fig. 12A only shows one mobility index, L/H

Fixed

350-351 – incomplete sentence

Fixed

352 – I think something is missing from the parenthetical note

Fixed

354 – reformulate to avoid use of contraction

Fixed

364 – citations needed or eliminate the reference to other scientists

Done

**Figures**

3 – panels C, D not called or discussed in text

We realize that not all of the panels were discussed in the text; however, we've opted to keep all four panels in the figure, which we have reorganized after our response to Reviewer 2 as these panels are consistent with InSAR result presentation common in the literature.

5 – what are the thin grey lines? How is velocity (panel B) calculated?  It appears somewhat different from what I expect based on the slope of the displacement measurements in A.  Also, are the points plotted at the date of the analyzed image each year?

The thin gray lines are the actual pixel displacement and velocity values. The velocity values were calculated using the central difference approximation. This is the most accurate way to perform a numerical derivative compared with forward differencing or backward differencing. The points are plotted on the same date as each image acquisition.

6 – replace "or" in first line of the caption with ", " if 3,668 m is the elevation of the top of the slide; revisit entire caption for spelling, capitalization.

Fixed

9 – the date of the slide is stated as June 29, in contrast with June 28 in the rest of the paper; I suggest adding something to state that the beginning of the hydrological year (0 on panel B x-axis) is not the same as the beginning of the calendar year

Fixed!

10 – colors did not come through for the version I received; caption states that the thick dashed line is limit of landslide material—is all the highlighted material in A the pre-collapse material or not?

The area shaded in gray represents the entire landslide deposit including material on its toe that did not fail on June 28$^{th}$. The dashed line represents the approximate lower limit of material that failed on June 28$^{th}$. We have added a supplementary figure in color and clarified the role of the dashed lines in the caption.

11 – define PDDS (both the acronym and how it is calculated); because snowmelt depends on both temperature and precipitation, it would be valuable to see annual precipitation plotted as well

PDDS is now defined in response to reviewer number 2 comments. I could not find an elegant way to display annual precipitation in a similar style to temperature; moreover, annual precipitation would confound both snowfall (accumulation) and rainfall (melting). It's not clear this would contribute to our overall analysis presented in Figure 11.

A1 – include geologic unit symbols

Figure A1 has now been moved to Figure 2 in the main text. More geologic symbols are included and properly defined in the caption.

I enjoyed reading the manuscript and think it will make a valuable contribution, and I encourage the authors to contact me with any questions or for clarification about the review.

Sincerely,

Alex Tye

alex.tye@utahtech.edu

We would like to thank the Reviewer for their comments that helped us make an even more robust manuscript and story regarding the Chaos Canyon landslide. We agree with the reviewer that this process will further aid us in understanding landslide processes in other alpine environments.

Best Wishes,
Matthew